# Deacetylation as a receptor-regulated direct activation switch for pannexin channels

Yu-Hsin Chiu [1,2,3 ✉], Christopher B. Medina[4], Catherine A. Doyle [1], Ming Zhou [5], Adishesh K. Narahari [1], Joanna K. Sandilos[1], Elizabeth C. Gonye [1], Hong-Yu Gao[2], Shih Yi Guo[3], Mahmut Parlak [4], Ulrike M. Lorenz[4], Thomas P. Conrads [5], Bimal N. Desai[1], Kodi S. Ravichandran[4] & Douglas A. Bayliss [1 ✉]

Activation of Pannexin 1 (PANX1) ion channels causes release of intercellular signaling molecules in a variety of (patho)physiological contexts. PANX1 can be activated by G protein-coupled receptors (GPCRs), including α1-adrenergic receptors (α1-ARs), but how receptor engagement leads to channel opening remains unclear. Here, we show that GPCR-mediated PANX1 activation can occur via channel deacetylation. We find that α1-AR-mediated activation of PANX1 channels requires Gαq but is independent of phospholipase C or intracellular calcium. Instead, α1-AR-mediated PANX1 activation involves RhoA, mammalian diaphanous (mDia)-related formin, and a cytosolic lysine deacetylase activated by mDia – histone deacetylase 6. HDAC6 associates with PANX1 and activates PANX1 channels, even in excised membrane patches, suggesting direct deacetylation of PANX1. Substitution of basally-acetylated intracellular lysine residues identified on PANX1 by mass spectrometry either prevents HDAC6-mediated activation (K140/409Q) or renders the channels constitutively active (K140R). These data define a non-canonical RhoA-mDia-HDAC6 signaling pathway for GαqPCR activation of PANX1 channels and uncover lysine acetylation-deacetylation as an ion channel silencing-activation mechanism.

[1] Department of Pharmacology, University of Virginia, Charlottesville, VA, USA. [2] Institute of Biotechnology, National Tsing Hua University, Hsinchu, Taiwan. [3] Department of Medical Science, National Tsing Hua University, Hsinchu, Taiwan. [4] Department of Microbiology, Immunology & Cancer Biology, University of Virginia, Charlottesville, VA, USA. [5] Inova Center for Personalized Health, Inova Schar Cancer Institute, Fairfax, VA, USA. ✉email: yhchiu@life.nthu.edu.tw; dab3y@virginia.edu

  **1**

Pannexin 1 (PANX1) forms oligomeric ion channels that are renowned for releasing nucleotides to mediate purinergic signaling in multiple (patho)physiological contexts[1–3]. Depending on the context, different channel activation mechanisms have been defined. During apoptosis, stepwise caspase-mediated cleavage of PANX1 causes a graded, but irreversible, channel activation to support release of ATP and other metabolites that attract phagocytes to facilitate immunologically silent clearance of dying cells and establish a local anti-inflammatory environment[4–7]. There is also evidence for direct, reversible PANX1 activation by physicochemical factors such as membrane stretch (e.g., in cancer metastasis)[8,9], high extracellular potassium (e.g., in seizures)[10–12], or elevated intracellular calcium (e.g., in aberrant cardiomyocyte depolarization)[13]. In addition, indirect activation of PANX1 by G protein-coupled receptors (GPCRs) has been implicated in numerous settings (e.g., injury-induced neuropathic pain, airway irritant responses, blood pressure regulation, etc.)[14–17]. However, the downstream signaling pathways and channel regulatory mechanisms that account for GPCR-mediated PANX1 activation have not been determined.

The GPCRs identified with PANX1 activation are typically associated with Gαq-containing heterotrimeric G proteins, and in some cases a role for intracellular $Ca^{2+}$ has been suggested. For example, Panx1 activation in Xenopus oocytes by co-expressed P2Y receptors could be mimicked with a calcium ionophore or direct exposure of Panx1-containing membrane patches to elevated $Ca^{2+}$[18]. However, a $Ca^{2+}$ contribution to receptor-mediated Panx1 activation was not directly tested, and a mechanism by which $Ca^{2+}$ might directly activate the channels was not defined. A $Ca^{2+}$-dependent mechanism was also implicated in PAR1-mediated activation of Panx1, for which concomitant actions of Rho-associated protein kinase (ROCK) were also required[14]; a specific mechanism by which ROCK contributed to Panx1 channel activation was not determined. It has also been suggested that phosphorylation of Tyr-198 of Panx1 might play a role in channel modulation by α1-AR[15], but recent observations indicate that Y198 phosphorylation of Panx1 is unaffected by α1-AR stimulation[19]. Thus, the receptor-mediated Panx1 activation mechanisms remain elusive.

In this work, using molecular and pharmacological tools, we show that Panx1 activation by the α1D-AR engages Gαq and an unconventional RhoA-mDia-HDAC6 signaling pathway to mediate direct activation of Panx1 channels by lysine deacetylation. The findings raise the possibility that acetylation–deacetylation may be a more general switch mechanism for ion channel activation. In addition, we demonstrate this α1-AR and HDAC6-mediated Panx1 activation in T lymphocytes, suggesting a potential signaling modality for neuro-immune modulation by GPCRs[20].

## Results

**PANX1 channels are activated by Gαq-coupled receptors.** We established a cellular reconstitution system in which co-expression of various Gαq-coupled receptors with either mouse or human PANX1 channels in HEK293T cells recapitulates receptor-mediated channel activation (Supplementary Fig. 1)[5,15]. We find that HEK293T cells have no detectable endogenous PANX1 currents (i.e., a null system), assessed using the channel blocker carbenoxolone (CBX)[5,15]; when transfected in those cells, mPanx1 generates basal whole-cell currents whereas hPANX1 is basally silent (Supplementary Fig. 1a–c). mPanx1 and hPANX1 are both activated by co-expressed α1D-adrenergic receptor (α1D-AR; Supplementary Fig. 1a–c), and mPanx1 is activated by multiple additional GαqPCRs (e.g., H1 histamine receptor, metabotropic glutamate receptor mGluR1; Supplementary Fig. 1e, f). Note that Panx1 channels are active over a wide voltage range and generate a weakly outwardly rectifying, open channel current (i.e., evident in I–V curves as large outward current at positive membrane potentials, with smaller, but clearly evident inward current at negative potentials)[5].

In α1D-AR-expressing cells, mPanx1 channel activation by the α1-selective agonist phenylephrine (PE) was observed in cell-attached patches (Fig. 1a, b)[5]. In this recording configuration, the agonist-stimulated receptor is physically separated from the Panx1 channels that are recorded within the membrane patch, thus implicating a diffusible signaling pathway. The α1D-evoked currents appear to be due to effects on channels already located on the cell membrane since prolonged agonist treatment did not appreciably alter cell-surface levels of the channel (Supplementary Fig. 1d). Importantly, α1D-mediated activation was fully intact in channels bearing a mutation in the PANX1 caspase cleavage site (Supplementary Fig. 1b, c), and it was reversible upon wash of PE (Supplementary Fig. 1c), indicating that it is independent of the irreversible cleavage-based channel activation mechanism we previously described[4–6].

In α1D-AR-expressing cells, whole-cell mPanx1 currents were activated by PE when recorded using pipettes containing GTP but not those containing GDPβS, a GDP analog that interferes with receptor-mediated Gα-GDP:GTP exchange (Fig. 2a–c). In addition, and as expected for this Gαq-coupled receptor, we found that PE-mediated stimulation of α1D-AR did not activate Panx1 channels expressed in fibroblasts derived from Gαq/11 double-knockout mice (Fq11 cells), but receptor-mediated channel activation was rescued when Gαq was re-expressed in those cells (Fig. 2d, e, g). Notably, receptor-mediated Panx1 activation in Fq11 cells was not rescued by co-expression of a mutated Gαq (A253K) construct (Fig. 2f, g) that is unable to couple to p63RhoGEF[21] but retains the ability to activate PLC (i.e., increase intracellular calcium) (Fig. 2h, i). This indicates that PLC activation is not sufficient for Panx1 activation; the lack of rescue

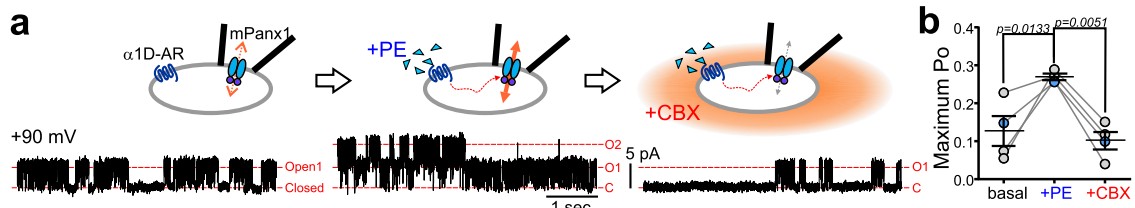

**Fig. 1 Panx1 channel activation by α1D-ARs requires a diffusible messenger. a** Exemplar cell-attached patch recording shows phenylephrine (PE, 20 µM)-induced activation of carbenoxolone (CBX, 50 µM)-sensitive channels in HEK293T cells expressing α1D-adrenergic receptors (α1D-ARs) and mPanx1. The schematic illustrates that, in this configuration, the liganded α1D receptor is physically separated from the recorded channel in the patch pipette, activation of which must involve a diffusible messenger. **b** Effects of PE and CBX on channel activity (maximum $P_O$) in those cell-attached patches ($n = 4$ membrane patches examined over 4 independent experiments). Data from the exemplar patch recording (in **a**) is represented by cyan dots. One-way ANOVA ($F_{2,9} = 11.35$, $p = 0.0035$) with Bonferroni's multiple comparisons test ($p$ values from comparisons are shown).

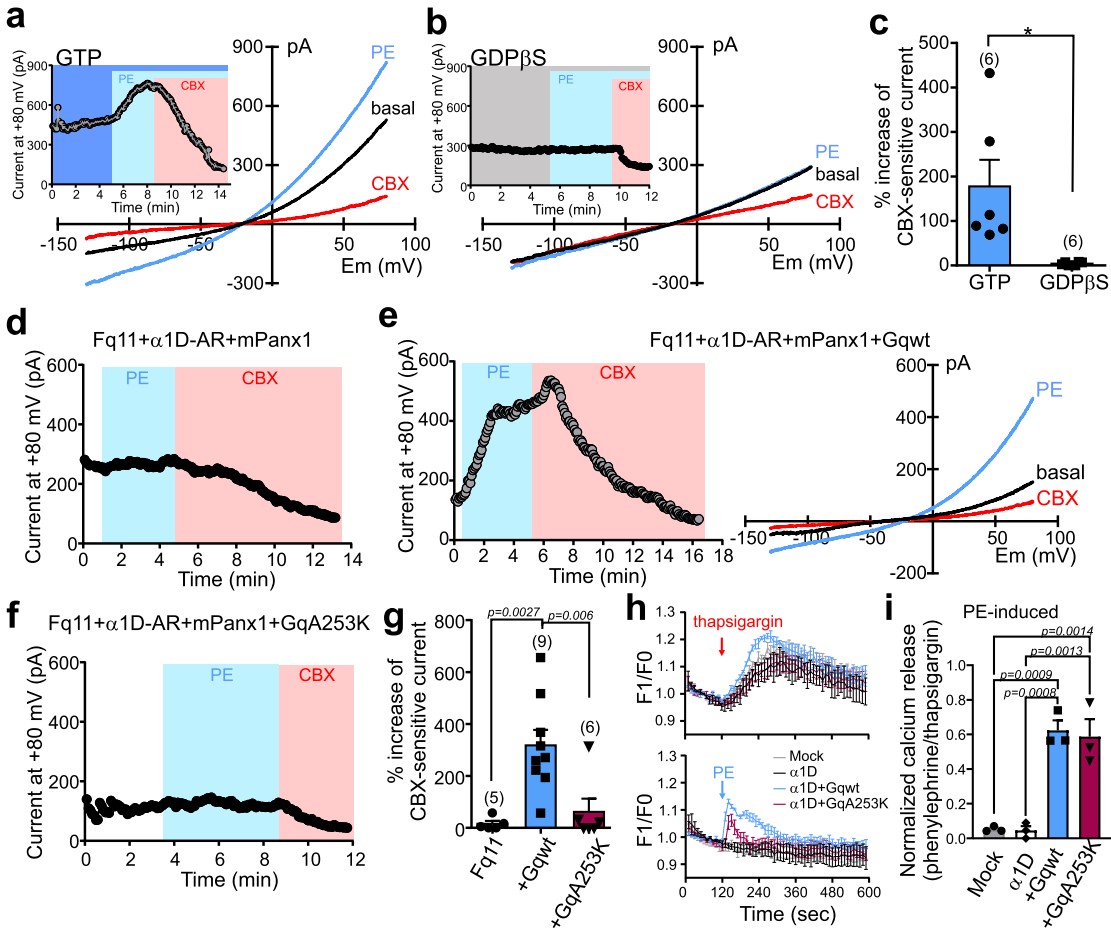

**Fig. 2 Gαq proteins are required for Panx1 channel activation by α1D-ARs. a–c** Effect of PE on whole-cell mPanx1 currents in α1D-AR/mPanx1-transfected cells recorded with either GTP (**a**) or GDPβS (**b**) in the pipette. *I*–V relationships depicted are from ramp voltage commands and time series from current amplitude at 80 mV (Insets). Grouped data (**c**) show that GDPβS diminished PE-induced current. PE-induced whole-cell current was normalized to basal CBX-sensitive current. *$p = 0.0160$ using two-tailed *t*-test. $n = 6$ cells per group examined over 3 independent experiments. **d–f** Effect of PE on whole-cell mPanx1 currents in fibroblasts from Gαq/11 double-knockout mice (Fq11) transfected with α1D-AR and mPanx1 alone (**d**), or when co-transfected with wild-type Gαq (**e**) or Gαq(A253K) (**f**). **g** Summary data reveal that only wild-type Gαq could rescue PE activation of mPanx1 in Fq11 cells. $n = 5$, 9, or 6 cells examined over 3 independent experiments. One-way ANOVA ($F_{2,5} = 11.63$, $p = 0.0009$) with Bonferroni's multiple comparisons test ($p$ values from comparisons are shown). **h** Intracellular calcium measurements in response to PE (20 μM) or thapsigargin (1 μM) in Fura-2AM loaded Gαq/11/12/13-deleted HEK293 cells transfected with α1D-AR, with or without different Gαq proteins ($n = 3$ biologically independent experiments). **i** Averaged peak signal for receptor-mediated calcium transients normalized to thapsigargin ($n = 3$ biologically independent experiments). One-way ANOVA ($F_{3,8} = 28.57$, $p = 0.0001$) with Bonferroni's multiple comparisons test ($p$ values from comparisons are shown). Summary data are presented throughout as mean ± s.e.m. PE phenylephrine.

with Gαq(A253K) suggests instead that there may be a role for Gαq-mediated Rho activation.

We also found that receptor-mediated Panx1 activation was largely unaffected when cells were recorded with pipettes containing high concentrations of BAPTA to rapidly chelate intracellular calcium (pCa~8), again consistent with the idea that a $Ca^{2+}$-dependent signaling pathway is dispensable for channel activation (Supplementary Fig. 2a, b). This is also in line with previous work reporting that PE-mediated ATP release from cells co-expressing α1D-AR and Panx1 was unaffected by intracellular calcium chelation[19]. Collectively, these results support a Gαq-mediated, diffusible signaling pathway interposed between α1D-AR and activation of Panx1 channels; this pathway is independent of PLC-$Ca^{2+}$ and may involve activation of RhoA.

**RhoA is necessary and sufficient for activation of PANX1 channels**. To examine whether RhoA is required for α1D-mediated Panx1 activation, we tested effects of two Rho inhibitors with different mechanisms of action: C3 exoenzyme, which ADP-

ribosylates and inactivates Rho family proteins[22]; and RhoA (T19N), which is a dominant-negative form of RhoA[23,24]. In HEK293T cells expressing α1D-AR and mPanx1, PE-activated whole-cell currents were strongly reduced when either C3 exoenzyme or RhoA(T19N) were co-transfected (Fig. 3a–c). To establish whether RhoA can act downstream of the α1D-AR receptor to activate Panx1 currents, we co-expressed a constitutively active RhoA(G14V) construct in hPANX1-expressing cells[23,25]. Strikingly, basal PANX1 currents were already elevated upon whole-cell access in cells co-expressing RhoA(G14V), and α1D-AR stimulation by PE did not further activate the current (Fig. 3d, g). This effect was not observed in control experiments using a RhoA(G14V) construct that was further mutated to disrupt interactions with all known RhoA effectors (Supplementary Fig. 2f)[23,25]; that is, in RhoA(G14V/T37Y)-expressing cells, PE evoked the characteristic increase in PANX1 current from low to undetectable baseline levels (Fig. 3e, g). These data indicate that RhoA activation is necessary and sufficient for α1D-mediated PANX1 activation.

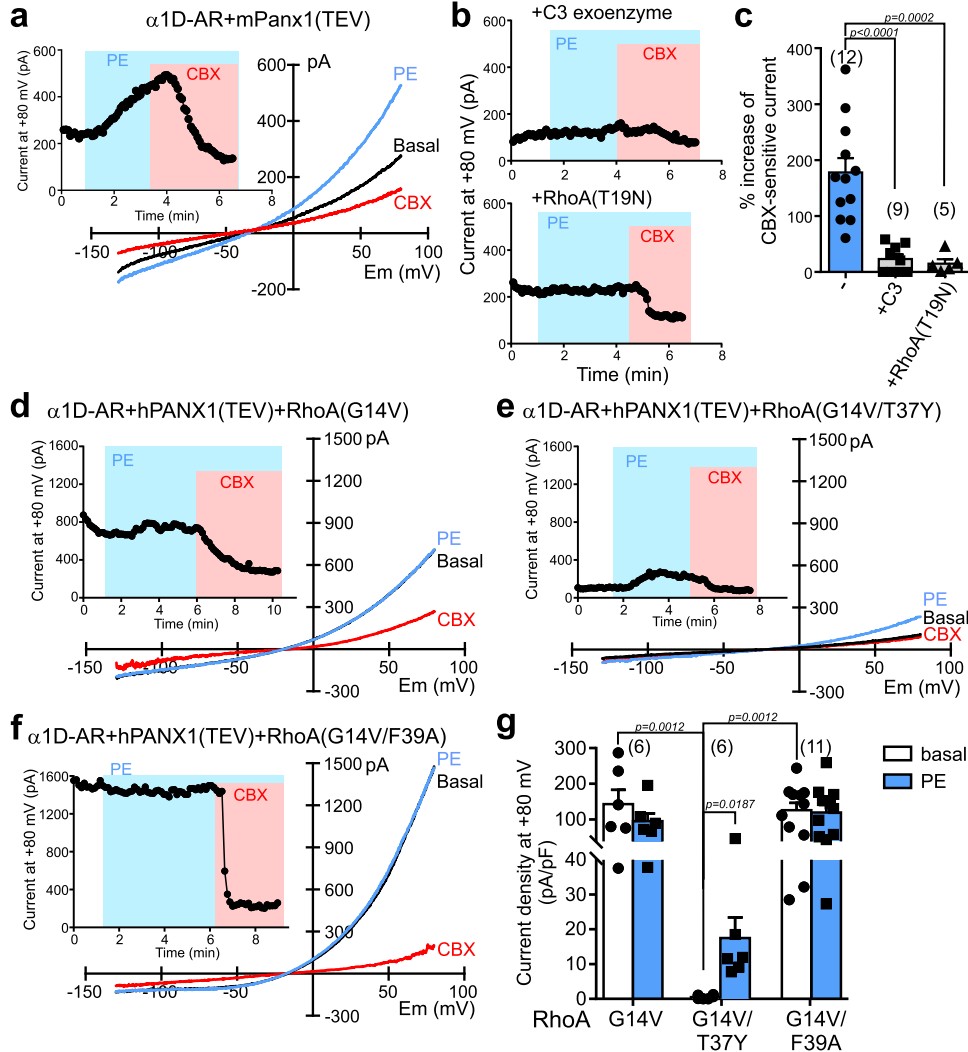

**Fig. 3 RhoA is necessary and sufficient for receptor-mediated PANX1 activation. a, b** Effect of PE on whole-cell currents in HEK293T cells transfected with α1D-AR and mPanx1(TEV) alone (**a**) or when co-transfected with either C3 exoenzyme or dominant-negative RhoA(T19N) (**b**). **c** Grouped data (mean ± s.e.m) show that overexpression of C3 exoenzyme or RhoA(T19N) reduced PE-evoked Panx1 current. $n = 12$, 9, or 5 cells per group examined over 4 independent experiments. One-way ANOVA ($F_{2,23} = 19.94$, $p < 0.0001$) with Bonferroni's multiple comparisons test. **d–f** Whole-cell currents in α1D-AR/hPANX1(TEV)-transfected cells co-expressing the constitutively-activated RhoA constructs: RhoA(G14V) (**d**), RhoA(G14V/T37Y), a control that cannot couple to downstream effectors (**e**), or RhoA(G14V/F39A) that only couples to mDia (**f**). **g** Summary data (mean ± s.e.m) reveal that the mDia-activating RhoA(G14V/F39A) construct can recapitulate increased basal hPANX1 currents and occluded PE effects observed with RhoA(G14V). $n = 6$, 6, or 11 cells per group examined over 5 independent experiments. Two-way ANOVA ($F_{2,20} = 7.776$, $p = 0.0032$) with Bonferroni's multiple comparisons test. PE phenylephrine.

Rho family GTPases have multiple downstream effectors, such as Rho-associated kinases (ROCK1, ROCK2), Src family kinases (SFKs), and mammalian diaphanous-related formins (mDia1~3)[23,26–28]. Activation of Panx1 channels by α1D-AR stimulation was unaffected by two chemically distinct ROCK inhibitors, Y27632 or H1152 (Supplementary Fig. 2c–e). This indicates that receptor-mediated channel activation can bypass ROCK, and implies involvement of a mechanism distinct from the Rho-ROCK-PANX1 pathway activated by hypotonic solution in airway epithelial cells[24]. Likewise, whereas SFKs are proposed to activate Panx1 channels by phosphorylation at Tyr-198 or Tyr-308[15,29], phenylalanine substitutions at those channel sites (Y198F/Y308F) did not disrupt Panx1 activation by α1D-AR stimulation (Supplementary Fig. 3). On the other hand, co-expression of RhoA(G14V/F39A), a variant of constitutively active RhoA that couples only to mDia (mDia1~3) (Supplementary Fig. 2f)[23,25], was able to recapitulate effects of RhoA(G14V)

in mimicking and occluding receptor-mediated PANX1 activation (Fig. 3f, g). Thus, we found large PANX1 currents under unstimulated conditions in cells transfected with RhoA(G14V/F39A), and those currents were not further increased by PE stimulation. These data suggest that RhoA stimulation of PANX1 is independent of ROCK or SFKs, and may instead involve mDia.

**mDia and HDAC6 mediate activation of PANX1 channels**. Under unstimulated conditions, mDia proteins are prevented from interacting with target proteins by an intramolecular interaction involving the so-called diaphanous autoregulatory domain (DAD)[30–32]; this autoinhibition is released by Rho-GTP binding to a GTPase binding domain (GBD) of mDia, and can also be undone experimentally by exogenous overexpression of a competing DAD mimetic peptide[32]. In cells expressing DAD, prominent basal PANX1 currents were observed under unstimulated conditions, and those currents were not further elevated

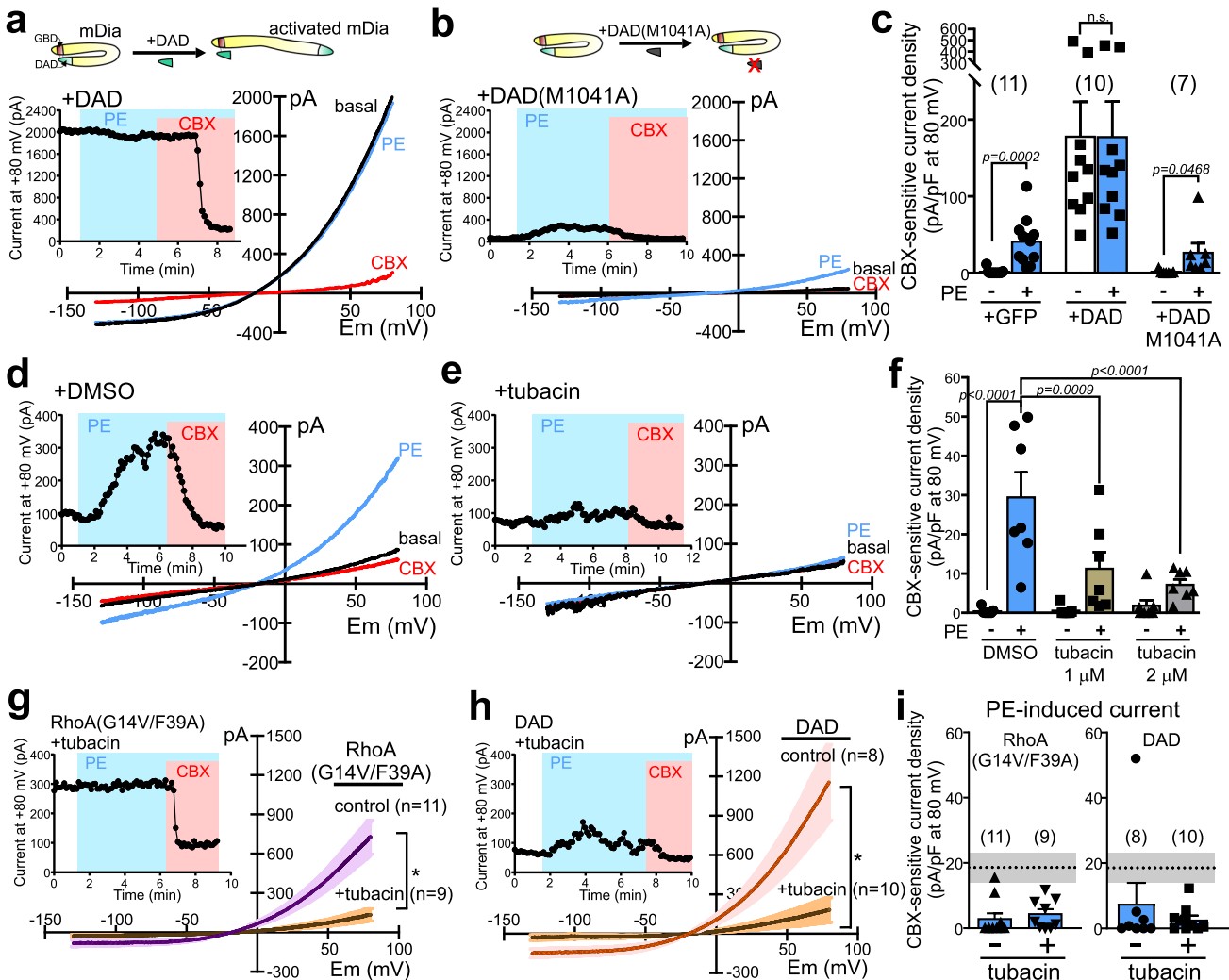

**Fig. 4 mDia and HDAC6 mediate α1D-AR activation of PANX1. a, b** Basal and PE-stimulated whole-cell currents in α1D-AR/hPANX1(TEV)-transfected HEK293T cells co-expressing a wild-type diaphanous autoregulatory domain (DAD) construct (**a**) or a mutated DAD(M1041A) version that cannot bind mDia (**b**). **c** Summary data (mean ± s.e.m) reveal that DAD, but not DAD(M1041A), increased basal PANX1 current and occluded PE effects. $n = 11$, 10, or 7 cells examined over 5 independent experiments. Two-way ANOVA ($F_{2,25} = 5.673$, $p = 0.0093$) with Holm-Sidak's multiple comparisons test ($p$ values from comparisons are shown). n.s. not significant. **d, e** Basal and PE-stimulated whole-cell currents in α1D-AR/hPANX1-transfected cells recorded with vehicle (DMSO, 14.1 μM) or tubacin (2 μM) in the pipette. **f** Concentration-dependent inhibition of PE-induced hPANX1 current by tubacin. $n = 7$ cells per group examined over 5 independent experiments. Data are presented as mean ± s.e.m. Two-way ANOVA ($F_{2,18} = 7.636$, $p = 0.0040$) with Bonferroni's multiple comparisons test. **g, h** Effects of tubacin (2 μM) treatment on basal whole-cell currents (mean ± s.e.m in α1D-AR/hPANX1(TEV)-transfected cells co-expressing the mDia-activating constructs, RhoA(G14V/F39A) (**g**) or DAD (**h**). Tubacin was added in culture media after transfection, and to the intracellular solution for whole-cell recording. Insets: representative time series of whole-cell current at 80 mV from α1D-AR/hPANX1(TEV)-transfected cells, with RhoA(G14V/F39A) or DAD, treated with tubacin. **i** Summary data (mean ± s.e.m) reveal that PE-induced CBX-sensitive currents from α1D-AR/hPANX1(TEV)-expressing cells were diminished by co-expression of RhoA(G14V/F39A) or DAD, regardless of tubacin treatment. $n = 11$, 9, 8, or 10 cells examined over 6 independent experiments. Dotted line and shaded area indicate mean and SEM of PE-evoked, CBX-sensitive current density from cells expressing α1D-AR and hPANX1. PE phenylephrine, CBX carbenoxolone.

by α1D-AR receptor stimulation (Fig. 4a, c). These effects of DAD were remarkably similar to those obtained with constitutively active RhoA(G14V), including the F39A variant that interacts only with mDia; they were not seen in control cells expressing a mutated DAD(M1041A) construct that cannot bind mDia (Fig. 4b, c)[32,33]. These data indicate that activated forms of RhoA, as well as DAD constructs that reproduce RhoA-mediated relief of mDia autoinhibition, can mimic and occlude receptor-mediated PANX1 activation.

HDAC6 is a cytosolic, class IIb histone deacetylase that mediates Rho- and mDia-induced microtubule deacetylation in osteoclast cells[34–36], suggesting that it could function as a downstream effector in α1D-AR-mediated Panx1 activation.

Consistent with this, PE-induced Panx1 current was inhibited by trichostatin A, a broad-spectrum class I and class II HDAC inhibitor (Supplementary Fig. 4a, b), and by tubacin, a more selective HDAC6 inhibitor (Fig. 4d–f)[37,38]. Moreover, tubacin treatment blocked the increase in basal current that was observed in cells expressing the mDia activators, RhoA(G14V/F39A) or DAD (Fig. 4g, h). Notably, despite the much lower initial current after tubacin, PE was still unable to activate Panx1 current (Fig. 4g–i). This suggests that the blunted α1D action in RhoA (G14V/F39A)- or DAD-expressing cells is not due to a "ceiling" effect but rather more likely represents convergence on a shared mechanism of action (i.e., occlusion). Together, these data suggest

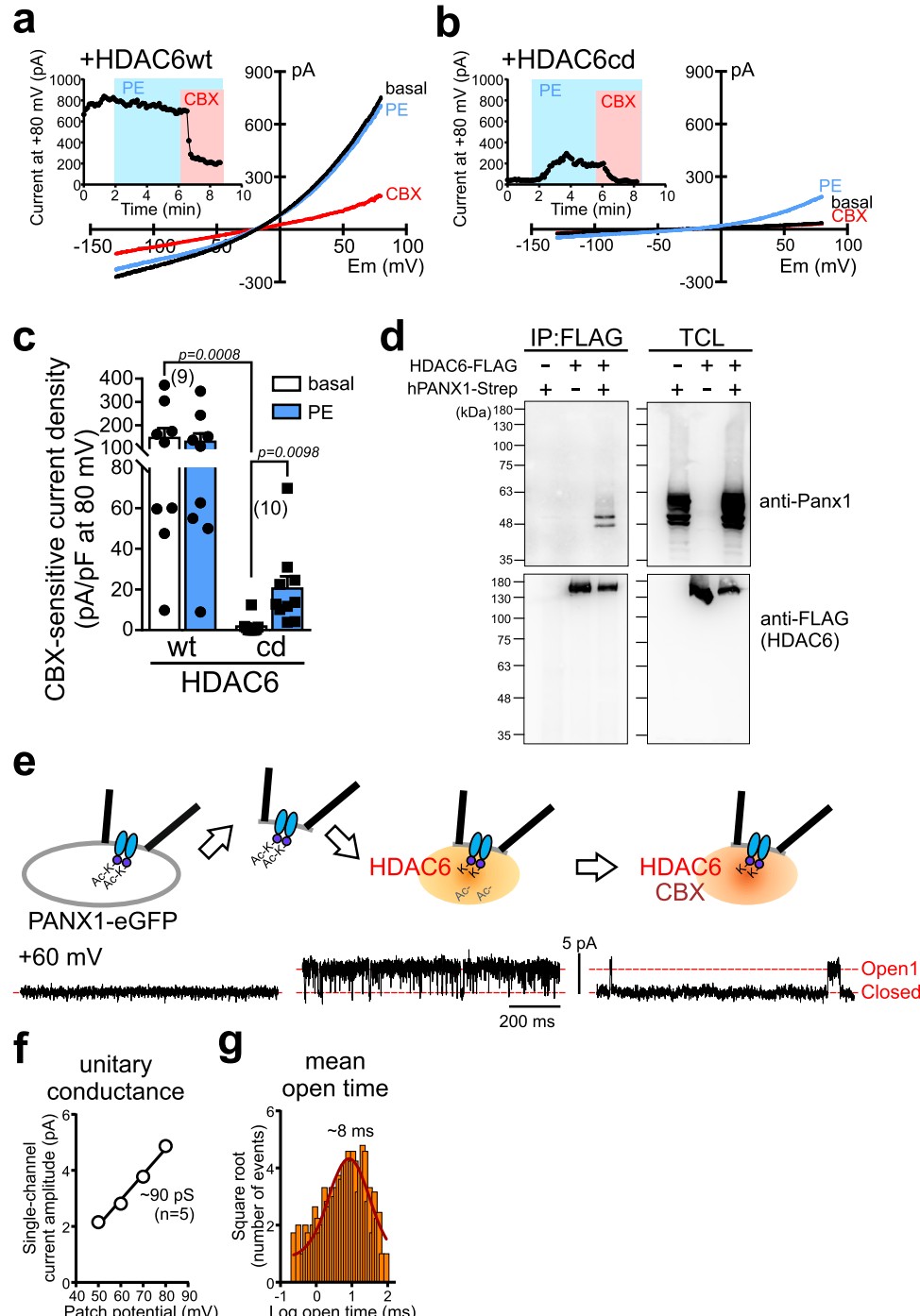

**Fig. 5 HDAC6 is sufficient to activate PANX1, even in cell-free conditions. a, b** Basal and PE-stimulated whole-cell currents in α1D-AR/hPANX1(TEV)-transfected HEK293T cells co-expressing wild-type (**a**) or catalytically inactive (**b**) forms of HDAC6. **c** Summary data (mean ± s.e.m) showing that the active form of HDAC6 increased basal PANX1 current and occluded PE effects. $n = 9$ or 10 cells examined over 3 independent experiments. Two-way ANOVA ($F_{1,17} = 18.07$, $p = 0.0005$) with Bonferroni's multiple comparisons test ($p$ values from comparisons are shown). **d** Co-immunoprecipitation showing interaction between HDAC6 and hPANX1 (from $n = 3$). FLAG-tagged HDAC6 and strep-tag-conjugated PANX1 were heterologously expressed in HEK293T cells. PANX1 proteins were co-immunoprecipitated with FLAG-tagged HDAC6. TCL total cell lysate. **e** Effect of HDAC6 and CBX on GFP-tagged hPANX1 channel recorded in inside-out patch configuration. **f, g** Averaged unitary I–V (**f**) and a representative open time distribution (**g**) for HDAC6-activated PANX1 channels in inside-out patches ($\gamma = 89.5 \pm 1.8$ pS, mean open time = 8.3 ± 1.3 ms, $n = 5$ independent membrane patches; mean ± s.e.m). PE phenylephrine, CBX carbenoxolone.

that HDAC6 may serve as an intermediary downstream of RhoA and mDia for α1D-AR-mediated PANX1 activation.

**HDAC6 directly activates PANX1 channels by deacetylation.** We next tested whether the cytosolic deacetylase HDAC6 can

activate PANX1. Heterologous expression of HDAC6, but not a catalytically inactive HDAC6, yielded large basal whole-cell PANX1 currents and occluded effects of α1D-AR activation (Fig. 5a–c), mimicking effects observed with either constitutively active RhoA or the DAD peptide. In addition, in a co-

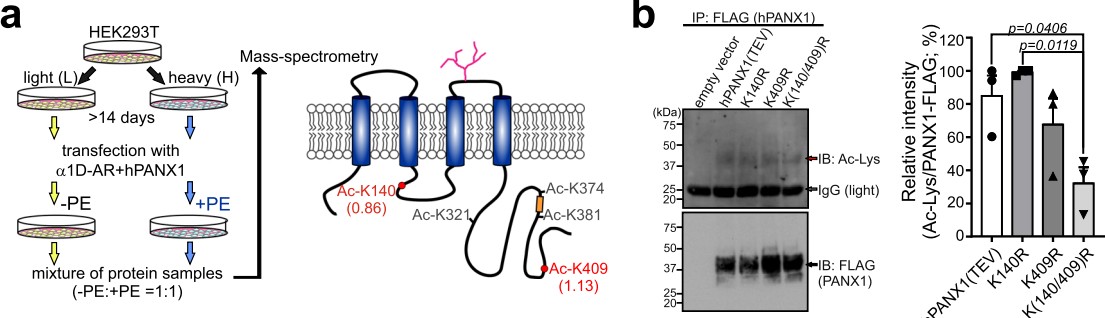

**Fig. 6 Identification of acetylated lysine sites on PANX1. a** SILAC/MS analysis of hPANX1-Strep immunoprecipitated from control or PE-treated α1D-AR/hPANX1-transfected HEK293T whole-cell lysates revealed multiple lysine acetylation sites on hPANX1 (from $n = 3$ biological replicates). Abundance levels for the acetylated forms of K140 or K409 in the cognate peptides were normalized to the median abundance of PANX1 and calculated as a ratio of PE-treated (H): vehicle-treated (L) for Lys-140 (0.86 ± 0.02; 0.88, 0.88, 0.83) and for Lys-409 (1.13 ± 0.71; 0.92, 0.02, 2.44). Other lysine sites (K321, K374, or K381) were inconsistently acetylated at baseline across the three samples and/or unaffected by PE treatment. **b** Example acetyl-lysine blot of immunoprecipitated FLAG-tagged and Arg-substituted hPANX1(TEV) constructs (left); acetyl-lysine signal was first normalized relative to FLAG, and then to peak acetyl-lysine signal for comparisons across experiments (right, $n = 3$ independent experiments). Data are presented as mean ± s.e.m. One-way ANOVA ($F_{3,8} = 6.735$, $p = 0.0140$) with Bonferroni's multiple comparisons test ($p$ values from comparisons are shown). PE phenylephrine.

immunoprecipitation assay, we found an interaction between co-expressed HDAC6 and PANX1 (Fig. 5d). These data support a hypothesis that HDAC6 may be able to directly activate PANX1 channels.

To test whether HDAC6 can act directly on PANX1 channels in a cell-free system, we excised inside-out membrane patches from HEK293T cells expressing PANX1. As reported previously[5], we found no basal PANX1 channel activity in this recording configuration under unstimulated conditions. Strikingly, however, when we applied purified HDAC6 to the cytosolic-facing aspect of the patch membrane there was a robust increase in channel activity, which was subsequently inhibited by carbenoxolone (CBX) (Fig. 5e). The HDAC6-activated channels presented with a unitary conductance of ~90 pS (Fig. 5f), as expected for PANX1 channels in inside-out patches[5]. Interestingly, the channels were less 'flickery' with longer open time (~8 ms; Fig. 5g) than receptor-activated PANX1 channels recorded in cell-attached configuration[5], likely reflecting more persistent or complete deacetylation when exposed continuously to HDAC6 in cell-free conditions. These data suggest that a deacetylation mechanism can directly mediate Panx1 activation.

To identify potential acetyl-lysine residues on the channel that may be involved in α1D-stimulated Panx1 activation, we performed mass spectrometry on Strep-tag-conjugated PANX1 immunoprecipitated from pooled whole-cell samples after stable isotope labeling by amino acids in cell culture (SILAC), with and without PE stimulation (Fig. 6a)[39]. From this analysis ($n = 3$), we tentatively identified several basally acetylated lysine residues on PANX1 (Lys-140, Lys-321, Lys-374, Lys-381, and Lys-409; Fig. 6a); among these, the acetylation at Lys-140 and Lys-409 was clearly detected from all three biological replicates, including those peptides with a C-terminally acetylated lysine (i.e., at the trypsin/Lys-C digestion site; Supplementary Fig. 4c–f; Supplementary Table 1). In addition, by acetyl-lysine immunoblot of immunoprecipitated FLAG-tagged PANX1, we found that arginine (R) substitutions at Lys-140 and Lys-409 resulted in reduced acetylation of the channel (Fig. 6b), consistent with constitutive acetylation on those PANX1 residues.

We next sought functional validation of a role for these particular residues in channel modulation. To test if deacetylation of Lys-140 and/or Lys-409 of PANX1 can evoke channel activation, we recorded from cells expressing two deacetylation-mimetic constructs, PANX1(K140R) or PANX1(K409R). Cells

expressing PANX1(K140R) presented with large basal whole-cell currents that could not be further activated by α1D-AR stimulation, whereas PANX1(K409R)-expressing cells had little-to-no basal current and normal PE-induced PANX1 currents (Fig. 7a–c). Notably, whereas Lys-140 is conserved in the mouse Panx1 channel, which can be activated by GαqPCRs (Supplementary Fig. 1a, e, f), Lys-409 is not conserved (i.e., Ala-409). These observations suggest a more prominent role for Lys-140 than Lys-409 in this channel activation mechanism.

In contrast to the K140R-substituted construct, PANX1 currents were not basally elevated and α1D-AR modulation was intact in cells expressing channels with Gln substitutions at either or both residues (Supplementary Fig. 5). These results with K→R and K→Q mutants suggest that constitutive channel activity and occlusion of α1D-AR effects observed with PANX1(K140R) requires the positive charge expected from a deacetylated Lys-140 (and as mimicked by R140). In addition, they also imply that alternative, non-HDAC6 mechanisms may contribute to receptor activation of these Gln-substituted channels. Nevertheless, the large basal PANX1 currents observed in cells expressing either DAD or HDAC6 were not observed with the PANX1-K(140/409)Q construct (Fig. 7d–g), indicating that their deacetylation-dependent effects require lysine residues at one or both of these sites. Moreover, basal activity of wild-type PANX1 in HEK293T cells co-expressing wild-type HDAC6 was significantly reduced by tubacin (Fig. 7h), further supporting the hypothesis that the deacetylase activity of HDAC6 is necessary for PANX1 channel activation. Importantly, the HDAC6-mediated increase in PANX1 channel activity in excised inside-out patches was not observed with a PANX1-K(140/409)Q mutant (Fig. 7i); these mutant channels were present and functional since they could be activated by C-terminal cleavage in the same membrane patches (Fig. 7j). Together, these data indicate that HDAC6 can activate PANX1 directly by deacetylation of Lys-140, and that dynamic acetylation–deacetylation of the channel provides a switch mechanism to regulate its activity.

**PANX1 channel activation by HDAC6 in T lymphocytes.** T lymphocytes express Panx1 channels and various GαqPCRs, including the α1-adrenoceptor[40]. We therefore determined effects of α1-AR activation on native Panx1 channel currents from T cells. PE induced a CBX-sensitive, Panx1-like current in freshly isolated mouse CD4-positive T cells (Fig. 8a, b, Supplementary Fig. 6a) and also activated human PANX1-like currents in a

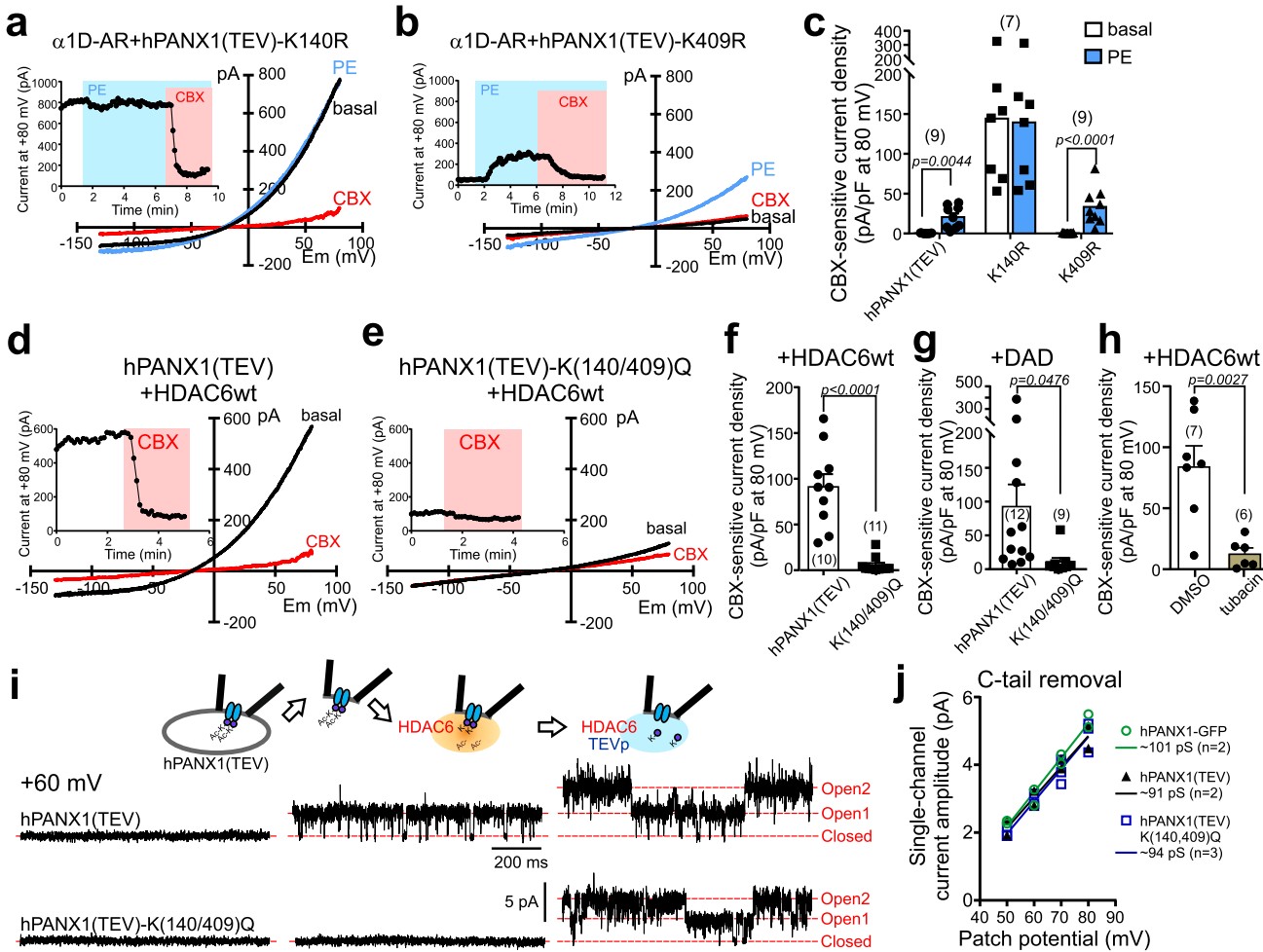

**Fig. 7 HDAC6-mediated channel activation by deacetylation of Lys-140 on PANX1. a, b** Basal and PE-stimulated whole-cell currents in α1D-AR-transfected HEK293T cells expressing the indicated Arg-substituted PANX1 constructs. **c** Summary data (mean ± s.e.m) reveal that Arg substitution at Lys-140, but not at Lys-409, increased basal PANX1 current and occluded PE effects. $n = 9$, 7, or 9 cells examined over 3 independent experiments. Two-way ANOVA ($F_{2,22} = 9.924$, $p = 0.0008$) with Bonferroni's multiple comparisons test ($p$ values from comparisons are shown). **d, e** Effect of HDAC6 on whole-cell current in HEK293T cells expressing hPANX1(TEV) or hPANX1(TEV)-K(140/409)Q. **f, g** Summary data reveal that HDAC6 (**f**) or DAD (**g**) did not activate basal whole-cell current in cells expressing Gln-substituted hPANX1(TEV) channels. $n = 10$ or 11 (**f**), 12 or 9 (**g**) cells examined over 6 independent experiments. Data are presented as mean ± s.e.m. Two-tailed unpaired $t$-test (**f** $t = 6.425$, $df = 19$; **g** $t = 2.118$, $df = 19$). **h** Grouped results showing tubacin (2 μM), but not DMSO, reduced basal whole-cell currents in cells expressing hPANX1(TEV) channels and wild-type HDAC6 proteins. $n = 7$ or 6 cells examined over 3 independent experiments. Data are presented as mean ± s.e.m. Two-tailed unpaired $t$-test ($t = 3.858$, $df = 11$). **i** Inside-out patch recordings of hPANX1(TEV) and hPANX1(TEV)-K(140/409)Q during exposure to HDAC6 and TEV protease. **j** Unitary $I$-Vs for each of the indicated PANX1 channel constructs after activation by C-terminal cleavage with caspase 3 or TEV protease; lines indicate fits by linear regression to estimate the unitary slope conductance (γ) for each group (green circles: hPANX1-eGFP, caspase: γ~101 pS; black triangles: hPANX1(TEV), TEVp: γ~91 pS; blue squares: hPANX1(TEV)-K(140/409)Q), TEVp: γ~94 pS; the $n$ values provided are for biologically independent patches). PE phenylephrine, TEVp Tobacco Etch Virus protease.

Jurkat T cell line (Fig. 8c, d). The PE-evoked current was not observed in Jurkat T cells after CRISPR/Cas9-mediated deletion of PANX1, indicating that it was indeed due to PANX1 (Supplementary Fig. 6b–d). To test whether HDAC6 can act directly on native PANX1 channels in T cells, we excised membrane patches from Jurkat cells and applied purified HDAC6 to the intracellular side of the membrane. As we had observed with the recombinant PANX1 channels, we found that HDAC6 was also able to activate native PANX1 channels in membrane patches excised from Jurkat cells (Fig. 8e, $n = 2$); the HDAC6-activated channel presented with a unitary conductance of ~86 pS, as expected for PANX1 (Fig. 8f, cf. Fig. 4f). These data indicate that GαqPCRs and HDAC6 activate native Panx1 channels in T lymphocytes.

## Discussion
PANX1 channels are reversibly activated by multiple Gαq-coupled receptors to mediate release of ATP and support intercellular purinergic signaling[15,16,18,41]. We examined cellular mechanisms by which GαqPCRs activate PANX1 channels, uncovering an unconventional signaling pathway linking the α1D-AR to PANX1 channels via RhoA-mDia-HDAC6, and defining a channel activation mechanism that involves a lysine acetylation–deacetylation switch on PANX1 (Fig. 8g).

### Lysine acetylation–deacetylation for ion channel modulation.
Protein lysine acetylation is a long-recognized form of reversible posttranslational modification most commonly associated with histones and transcriptional regulation; however, other roles for

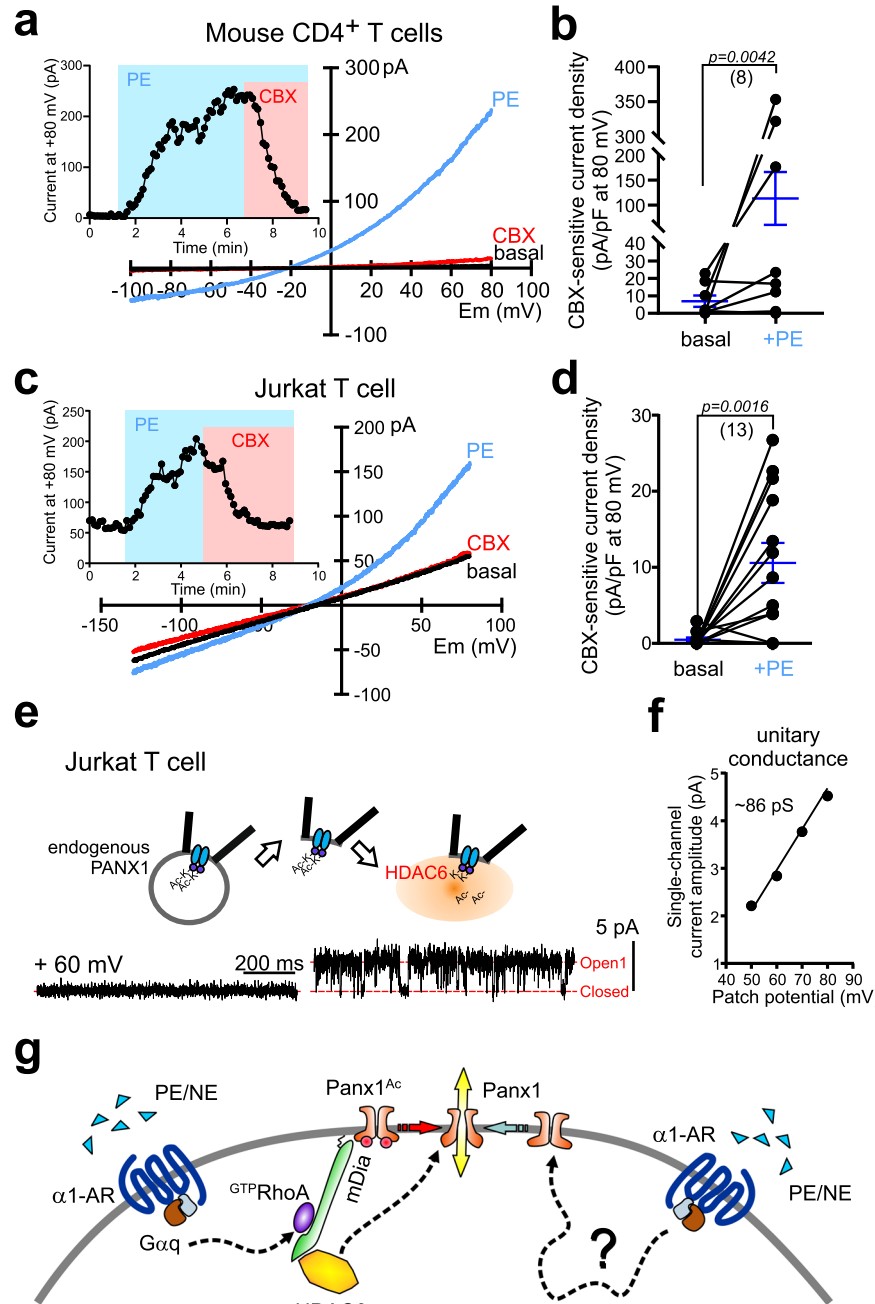

**Fig. 8 HDAC6 activates native Panx1 channels in T lymphocytes. a** Exemplar recording from a freshly isolated mouse CD4+ T cell that responded to PE (20 μM) with an increase in whole-cell CBX-sensitive current and *I*–V characteristics of Panx1. **b** Summary data (mean ± s.e.m, blue lines) showing CBX-sensitive whole-cell current density in mouse CD4 + naïve T cells, before and after bath application of 20 μM PE ($n = 8$; 5 $T_{reg}$ and 3 $T_{eff}$ cells examined over 3 independent experiments); one-tailed paired *t*-test (t = 2.014, df = 7). **c** Representative whole-cell recording from Jurkat T cell that showed increased CBX-sensitive PANX1-like current in response to PE (20 μM). **d** Summary data (mean ± s.e.m, blue lines) showing CBX-sensitive currents before and after PE stimulation on individual Jurkat T cells; one-tailed paired *t*-test ($n = 13$ cells examined over 3 independent experiments; t = 3.682, df = 12). **e, f** Inside-out recording from a Jurkat T cell showing activation by HDAC6 of a PANX1-like channel (**e**), with corresponding unitary *I*-V (**f**). **g** Schematic illustrates the RhoA-mDia-HDAC6 signaling pathway interposed between the Gαq-coupled α1-AR and Panx1 channel activation by lysine deacetylation and alternative pathway(s) not yet defined. PE phenylephrine, NE norepinephrine, CBX carbenoxolone.

protein acetylation are increasingly being recognized[36]. Notably, HDAC6 is found in the cytoplasm where its best known target is tubulin, deacetylation of which can lead to microtubule destabilization and changes in mechanical stress on cell membranes[34,35,42]. Although PANX1 is reported to be a stretch-activated channel[9], it is unlikely that HDAC6-mediated PANX1 activation was secondary to mechanical effects of tubulin deacetylation, especially in excised patches where such physical forces

would be minimal and channel activation by HDAC6 depended on specific lysine residues on PANX1. It is also worth mentioning that HDAC6 modulates various metabolic and signaling enzymes, and lysine deacetylation can be a prelude for protein ubiquitylation and proteasomal degradation[35]. Indeed, the only other known channel regulatory mechanisms associated with direct acetylation–deacetylation involve effects on channel stability (e.g., channel acetylation inhibited ubiquitylation of epithelial Na+

channels and Connexin32)[43,44] or membrane localization (e.g., channel deacetylation enhanced membrane trafficking of cardiac Na+ channels, Na$_V$1.5)[45]. However, we found no effects on membrane PANX1 levels even after extended agonist treatment in cells and, again, those metabolic actions are unlikely to account for increased PANX1 channel activity evoked by HDAC6 in cell-free membrane patches.

Interestingly, the direct PANX1 channel activation by lysine modification that we revealed is loosely analogous to a sumoylation-based regulation first discovered for K2P1 background potassium channels[46]. In that case, previously inactive channels at the cell membrane were unsilenced by enzymatic removal of small ubiquitin-like modifier (SUMO) protein adducts, rather than acetyl groups, from a specific channel lysine residue[46]; however, a signaling pathway that accesses the sumoylation process for K2P1 channel regulation has not been identified, and K2P1 activation does not depend on a specific charge at the key lysine residue[46]. By contrast, the lysine deacetylation of PANX1 channels we describe is triggered by a receptor-activated RhoA-mDia-HDAC6 pathway, and channel activation could be mimicked by replacement of Lys-140 with another positively charged residue in PANX1-K140R but not by the neutral substitution obtained with PANX1-K140Q. This suggests that a positive charge at that site (either Arg or deacetylated lysine) may be sufficient for constitutive activation of PANX1 channels. It is important to point out that although Gln substitutions on PANX1 did not induce basal currents and interfered with channel activation by HDAC6, an α1D-activated PANX1 current could still be evoked. This implies that those lysine residues are required for deacetylation-dependent channel activation, but not for an alternative form of GαqPCR-mediated activation (Fig. 8g). It remains to be determined whether and how electrostatic mechanisms supported by Lys-140 might contribute to basal PANX1 channel activation, and what alternative signaling processes support receptor activation of the unacetylated, but basally silent PANX1-K(140/409)Q channels.

**Receptor-activated signaling pathways for PANX1 regulation.** Our experiments have mapped a mechanism that involves signaling from α1D-AR to PANX1 via RhoA, mDia, and HDAC6. Even as this overall signaling pathway has not been previously delineated, the connections between individual molecular constituents in the pathway have been described. It is well established that p63RhoGEF can transduce Gαq receptor stimulation to RhoA activation[21,47]. Consistent with this, we find that a Gαq construct that does not interact with p63RhoGEF (unlike wild-type Gαq) is unable to rescue PANX1 channel activation in Gαq/11-deleted cells. Likewise, mDia is a well-known downstream effector for RhoA signaling, typically involved in mediating the actin nucleation and elongation associated with Rho-dependent cytoskeletal rearrangements[25,27,31]. In the context of RhoA-dependent PANX1 channel activation, a role for mDia as the relevant RhoA effector was supported by the observation that an F39A variant of the constitutively active RhoA(G14V) construct that couples only to mDia was sufficient for PANX1 channel activation. More limited information links RhoA and mDia to HDAC6, but this was implicated in the context of RhoA-mediated disruption of osteoclast maturation that involves mDia2 activation of HDAC6, and subsequent tubulin deacetylation[34]. In the present work, we provide evidence that PANX1 is itself a substrate for HDAC6, and link this channel modulation to receptor, RhoA and mDia signaling using pharmacological inhibitors of HDAC6 and mutagenesis of the relevant acetylated lysine residues on PANX1.

Given the widespread importance of purinergic signaling and the ability of activated PANX1 channels to mediate release of ATP and other metabolites, there has been much interest in understanding PANX1 regulation in different contexts. Collectively, these studies have revealed a number of surprising activation mechanisms for this remarkable channel[1,48]. This includes the first example of an irreversible proteolytic activation mechanism by C-terminal caspase cleavage of PANX1[4,6], which has subsequently been observed in other channels[49]. In another example, metabotropic actions by the NMDA receptor channel and the type I TNF receptor appear to regulate PANX1 by a Src family kinase-mediated tyrosine phosphorylation mechanism[29,50,51]. Likewise, our present examination of PANX1 channel activation by Gαq-coupled receptors reveals yet another unexpected signaling pathway, involving RhoA-mDia and an HDAC6-dependent channel deacetylation, and provides additional potential targets for regulating PANX1-dependent intercellular signaling.

**Receptor- and HDAC6-mediated regulation of T lymphocytes.** We previously showed that PANX1 channels are irreversibly activated in apoptotic T lymphocytes to release nucleotide find-me signals and other metabolites that expedite corpse clearance and support an anti-inflammatory environment[4,7]. Whether GPCR ligands activate PANX1 in T cells has never been explored despite a growing realization that various classical neurotransmitters, including norepinephrine, may serve as modulators of immune cell function[20]. We now provide functional evidence for activation of PANX1 channels by α1-AR signaling in CD4+ T cells, raising the possibility that α1-AR-activated PANX1 channels contribute to sympathetic regulation of immune response. A sympathetic neuronal-immune interaction is well recognized in the cholinergic anti-inflammatory pathway in which norepinephrine or epinephrine induces acetylcholine release from T cells to dampen macrophage activation[52–54]. Whereas those anti-inflammatory effects are attributed primarily to β2-adrenoceptors, it was reported that α1-AR signaling in T cells modulates splenic secretion of interferon γ in an arthritis model[55]. In addition, our finding that HDAC6 can directly activate PANX1 channels in T cells suggests other pathways for immune regulation. For example, HDAC6 and PANX1 are translocated to sites of contact with antigen presenting cells, where they contribute to the organization of immune synapse and T cell activation[56,57]. Conversely, inhibition of HDAC6 or PANX1 attenuates disease progression in some inflammatory and/or autoimmune mouse models (e.g., arthritis, multiple sclerosis)[58–60], and HDAC6 negatively regulates activity of CD4+ Foxp3+ regulatory T cells[61]. Our experiments showing activation of PANX1 channels by multiple GαqPCRs, some of which are also found on T cells and implicated in immune modulation (i.e., 5-HT2, H1, mGluR1)[20], suggests the exciting possibility that receptor/HDAC6-activated PANX1 channels could contribute to neurotransmitter-mediated immune modulation.

## Methods

**Animals and reagents.** Mice (C57BL6/J) were obtained from Jackson Labs and used following procedures adhering to National Institutes of Health Animal Care and Use Guidelines and approved by the Animal Care and Use Committee of the University of Virginia. Mice were housed in 12 h light/12 h dark cycle at room temperature with 40–70% humidity, with access to food and water ad libitum. All chemicals were purchased from Sigma–Aldrich unless otherwise stated. EZ-link-sulfo-NHS-LC-biotin and Fura-2AM were obtained from Thermo Fisher Scientific. Recombinant human HDAC6 was purchased from Millipore. A pro-caspase 3 Δ28/175TS deletion mutant in pET-22b (+) (Novagen) was purified from E.coli BL21 (DE3) cells and activated by thrombin cleavage, and Tobacco Etch Virus protease (TEVp, in pRK793) was purified from E. coli BL21(DE3) CodonPlus-RIL cells (Stragene)[5].

**Plasmids and cloning**. Plasmid containing mouse adrenergic α1D receptor was obtained from OriGene (MR222643) as previously reported[15,62]. Histamine H1 receptor and metabotropic mGluR1 receptor were obtained from Drs. Graeme Milligan (University of Glasgow)[63] and Stephen Ikeda (NIAAA, NIH)[64]. Wild-type Gq construct (Gqwt) and GqA253K construct was a gift from Dr. John Tesmer (Purdue University, IN)[21]. C3 exoenzyme and RhoA constructs (T19N, G14V, G14V/T37Y, G14V/F39A) were gifts from Dr. Kodi Ravichandran (University of Virginia, VA)[23]. pEGFPm-DAD and pEGFPm-DAD(M1041A) were gifts from Dr. Arthur Alberts (Addgene #25410, #25411)[32]. pcDNA-HDAC6-FLAG (Addgene #30482) and pEGFP-N1-HDAC6cd (catalytic-deficient HDAC6 carrying H216A and H611A mutations; Addgene #36189) were gifts from Dr. Tso-Pang Yao[65,66]. To generate eGFP-tagged wild-type HDAC6, catalytic-deficient HDAC6 was removed from pEGFP-N1-HDAC6cd by enzyme digestion using BglII and HindIII, and was replaced by wild-type HDAC6. The wild-type HDAC6 DNA was amplified by polymerase chain reaction (PCR) from pcDNA-HDAC6-FLAG, using GoPro DNA polymerase (Promega; primers provided in Supplementary Table 2), digested using BglII and HindIII and ligated with T4 DNA ligase (New England Biolabs).

Plasmids encoding mouse Pannexin 1 (mPanx1) and human Pannexin 1 (wild-type hPANX1, hPANX1(TEV), or C-terminally eGFP-tagged hPANX1) were previously described[5,6,15]. pFastBac containing C-terminally Strep-tagged hPANX1 was kindly provided by Dr. Mark Yeager (University of Virginia, VA). To generate lysine-to-arginine or lysine-to-glutamine mutations (K140Q, K409Q, K140Q/K409Q, K140R, and K409R) of human PANX1, site-directed mutagenesis was performed on hPANX1(TEV) construct using Pfu Turbo polymerase (Agilent) and the combinations of primers listed in Supplementary Table 2.

Plasmid encoding Tet-on spCas9 was modified from pCW-Cas9[67] (provided by Drs. Eric Lander and David Sabatini; Addgene #50661) to replace the puromycin-resistance gene with blasticidin resistance marker. pLX-sgRNA-BfuAI-2k plasmid (Addgene #112915, a gift from Dr. Ren-Jang Lin)[68] was engineered to encode two different small guide RNAs (sgRNAs) against human PANX1 gene. First, the blasticidin resistance gene of pLX-sgRNA-BfuAI-2k was replaced by zeocin resistance marker, and one additional U6 promoter was subcloned into this plasmid (pLX-2×sgRNA). Complementary oligonucleotides encoding two sgRNAs (sgRNA 1: GCCTTCACCCAGTCACCGGC; sgRNA 2: GATGGTCACGTGCATTGCGGT) were subsequently inserted into pLX−2× sgRNA using T4 DNA ligase at BsbI and BsmMI sites.

All constructs were verified by DNA sequencing.

**Mammalian cell culture and transfection**. HEK293T cells (passage 7–20, ATCC), embryonic fibroblasts (Fq11 cells) derived from Gq/11 knockout mice[69], and Gq/11/12/13-deleted HEK293 cells[65,66] were cultured at 37 °C with humidified air containing 5% $CO_2$ in Dulbecco's Modified Eagle Medium (DMEM, high glucose, Gibco) containing 10% fetal bovine serum (FBS, Gibco), penicillin, streptomycin, and sodium pyruvate. Fq11 cells were kindly provided by Dr. Patricia Hinkle (University of Rochester, Rochester, NY)[69], and Gq/11/12/13-deleted HEK293 cell[70] was a generously gift from Dr. Asuka Inoue (Tohoku University, Miyagi, Japan). Jurkat T cells (E6.1, ATCC) were maintained in RPMI 1640 (Corning) with 10% FBS, penicillin, streptomycin, and L-glutamine at a density of 0.5–1.5 cells ml$^{-1}$. Transfections were carried out by using either Lipofectamine2000™ (Invitrogen) or Transporter™ 5 (Polysciences) following manufacturers' manuals.

**Deletion of PANX1 from Cas9 stably-expressed Jurkat T cell line**. For production of lentivirus expressing Tet-on spCas9, HEK293T cells were seeded in 10 cm dish, and transfected the following day with pMD2.G and psPAX packaging vectors (Addgene #12259, Addgene#12260; both from Dr. Didier Trono), along with the targeting construct encoding Tet-on spCas9, using FuGENE-6 (Promega). Media was refreshed 6 h after transfection. Viral supernatants were collected 48 h and 72 h after first media refreshment, flash-frozen with liquid nitrogen, and stored at −80 °C. For viral transduction, Jurkat T cells (1 ml, $5 \times 10^5$ cells ml$^{-1}$) were incubated with 3 ml viral supernatant in a six-well plate with protamine sulfate (10 µg ml$^{-1}$; from 10 mg ml$^{-1}$ stock). This mixture was spun $1000 \times g$ at 32 °C for 2 h; this procedure was repeated the following day. After 24 h, virally transduced Jurkat T cells were cultured in media containing blasticidin (15 µg ml$^{-1}$) and zeocin (200 µg ml$^{-1}$) for 3 and 5 days, respectively. Virally transduced Jurkat T cells were treated with doxycycline (50 µM) for 48 h to induce expression of spCas9. After 48 h of induction, $5–10 \times 10^6$ cells were washed and resuspended in 400 µl plain RPMI 1640, followed by electroporation at 250 mV for 25 ms per pulse (BTX Electro Square Porator T820, Harvard Apparatus) with 10 µg of plasmids containing two PANX1 sgRNAs. Electroporated cells were further incubated in doxycycline for another 48 h, and then were cultured in selection media containing zeocin (400 µg mL$^{-1}$). After 1 week, cells were serially diluted into 96-well plates to generate oligo clones and then transferred to larger wells for continued expansion.

**Isolation of mouse primary CD4$^+$ T cells**. Naïve CD4$^+$ T cells were isolated from spleens of wild-type C57BL/6 J mice (8–14 weeks; both sexes). In brief, dissected mouse spleens were gently ground through 70 µm cell strainer mesh (Falcon), and naïve CD4$^+$/CD25$^+$ (regulatory; T$_{reg}$) or CD4$^+$/CD25$^-$ (effector; T$_{eff}$) T cells were isolated and collected using a CD4$^+$CD25$^+$ T cell isolation kit (Miltenyi Biotec;

#130-091-041) and magnetic separation columns (Miltenyi Biotec, #130-042-201 and #130-042-401) according to manufacturer's protocol. Purity of T cell populations were verified by using flow cytometry (Supplementary Fig. 6a; CD3-PerCP-Cy5.5 1:100, eBioscience, #45-0031-80; CD4-Pacific Blue 1:100, eBioscience, #57-0042-82; and Foxp3-APC 1:100, eBioscience, #17-5773-80; FlowJo v.10.6.2). Isolated CD4$^+$ T cells were resuspended in RPMI 1640 containing 10% FBS, 1% PSQ, 1% sodium pyruvate, 1% HEPES, and 1% nonessential amino acids, and plated on poly-L-lysine coated coverslips >1 h before recording. Whole-cell recording of CD4$^+$ T cells was carried out within 48 h after isolation.

**Electrophysiology**. All voltage-clamp recordings were carried out at room temperature, using an Axopatch 200B amplifier controlled by pCLAMP10 software and digitized using Digidata 1322 A digitizer (all Molecular Devices). Micropipettes were pulled from thin-walled, fire-polished borosilicate glass capillaries (Harvard Apparatus) using a P-97 Flaming/Brown Micropipette Puller (Sutter Instrument), and coated with Sylgard 184 silicone elastomer (Dow Corning Corporation).

Cell-attached patch recordings were performed using micropipettes with resistance of 5–10 MΩ, and filtered to 1 kHz using the Axopatch 200B. Bath and pipette solution contained 140 mM NaCl, 3 mM KCl, 2 mM $MgCl_2$, 2 mM $CaCl_2$, 10 mM HEPES, and 10 mM glucose (pH 7.3; ~300 mOsm). After ≥10 GΩ seals were obtained, cells were exposed to bath solution containing phenylephrine (PE, 20 µM). After steady-state channel activity was acquired (at +90 mV), the bath solution was further exchanged for one containing both PE (20 µM) and carbenoxolone (CBX, 50 µM). Maximum Po is derived from NPo divided by the maximum numbers of channel observed in a given patch.

Whole-cell recordings were obtained in the same HEPES-based bath solution (above) using 3–5 MΩ borosilicate glass patch pipettes, filled with internal (pipette) solution composed of: 100 mM CsMeSO$_4$, 30 mM TEACl, 4 mM NaCl, 1 mM $MgCl_2$, 0.5 mM $CaCl_2$, 10 mM HEPES, 10 mM EGTA, 3 mM ATP-Mg, and 0.3 mM GTP-Tris (pH 7.3; ~290 mOsm). In some experiments, intracellular free calcium was buffered to 10 nM by replacing EGTA with BAPTA (tetrapotassium salt, 14.3 mM), or GTP-Tris was replaced with GTP-lithium or GDPβS-lithium (also 0.3 mM). Ramp voltage-clamp commands were applied at 7-s intervals, with voltage ranging from −130 to 80 mV (0.2 mV/ms) in HEK293T and Jurkat T cells or from −100 to 80 mV (0.26 mV/ms) in mouse CD4$^+$ T cells. CBX-sensitive current was taken as the difference in current at +80 mV before and after CBX application, and was normalized to cell capacitance (i.e., current density). PE-induced current was determined by the difference in CBX-sensitive current at +80 mV, before and after PE application in bath. To ensure that recorded cells co-expressed signaling proteins (e.g., RhoA(T19N) or C3 exoenzyme) along with α1D-ARs and mPanx1 channels, we transfected cells with a DNA mixture that included (from highest to lowest concentration) α1D-AR, signaling protein constructs, mPanx1 channel, and a GFP-containing vector. Thus, co-transfected cells were identified by expression of GFP and presence of basal outwardly rectifying currents that were sensitive to CBX (i.e., Panx1 currents), the components that were transfected at the lowest DNA concentrations.

Inside-out patch recording were obtained using micropipettes with resistance of 5–10 MΩ; data were filtered to 5 kHz using an 8-pole low-pass Bessel filter (LPF-8, Warner Instruments) and digitized at 20 kHz. Pipettes were filled with HEPES-based bath solution (above) and, after seal formation (≥10 GΩ), the membrane patch was excised and the bath solution was exchanged for an inside-out solution composed of: 150 mM CsCl, 5 mM EGTA, 10 mM HEPES, and 1 mM $MgCl_2$ (pH 7.3). Patches were held at +50 to +80 mV (Δ 10 mV). We excluded membrane patches that displayed non-Panx1 channel activity, as determined by lack of CBX sensitivity and unitary channel properties unlike Panx1 channels[5]. As previously reported, we did not observe basally-active Panx1 channels immediately following patch excision[5]. Recombinant human HDAC6 (Millipore) was applied near the patch under stop-flow conditions to a final concentration of ~1 µg ml$^{-1}$. Whenever possible, after steady-state channel activity was obtained (usually within 8 min after HDAC6 application), activated caspase 3 or TEV protease was applied (~1–2 µg ml$^{-1}$) and the steady-state channel activity was recorded[5]. Single-channel recordings were analyzed using pCLAMP10; channel data were filtered to 2 kHz using an 8-pole low-pass Bessel filter for analysis of open probability, and filtered to 1 kHz for presentation.

**Cell-surface biotinylation and immunoprecipitation**. Cell-surface biotinylation were carried out in HEK293T cells that were transiently transfected with wild-type PANX1 and FLAG-tagged α1D-AR for ~24 h using Transporter™ 5 (Polysciences). Transfected cells were washed and incubated in HEPES-based bath solution (above) for 15 min at room temperature, and subsequently incubated with the same buffer containing 20 µM PE for additional 0~20 min (Δ 5 min). After PE stimulation, these cells were washed with cold phosphate-buffered saline (PBS, Gibco) and incubated in cold PBS containing EZ-Link Sulfo-NHS-Biotin™ (1 mg ml$^{-1}$; Thermo Scientific) for >1 h at 4 °C with gentle rocking. After biotin-containing PBS was removed, cells were further incubated with PBS containing 100 mM glycine at 4 °C for >30 min to quench the reaction. Cells were lysed in PBS buffer containing 1% Triton X-100, protease inhibitor cocktail (Sigma–Aldrich; P8340), 10 mM NaF, and 10 mM NaVO$_3$ (lysis buffer). Protein samples were mixed with streptavidin-agarose beads (Thermo Scientific) for >2 h at 4 °C to collect biotinylated proteins.

To determine constitutive lysine acetylation on PANX1, C-terminally FLAG-tagged hPANX1(TEV) constructs, with or without mutation of Lys-140 or Lys-409, were transiently transfected in HEK293T cells using Transporter™ 5. Cells were incubated with 2 μM tubacin after transfection (24 hr), transferred to lysis buffer containing 5 μM tubacin and 2 μM trichostatin A (TSA). Protein samples were mixed with anti-FLAG M2 agarose beads (Sigma–Aldrich, #A2220) for >2 h at 4 °C, with constant shaking.

To examine protein–protein interaction between PANX1 and HDAC6, HEK293T cells were transiently transfected with Strep-tagged PANX1 and FLAG-tagged HDAC6 for ~24 h using Transporter™ 5. Cells were lysed and protein samples were prepared as mentioned above. Anti-FLAG M2 affinity agarose gel (Sigma–Aldrich) was used to pull down HDAC6-FLAG proteins.

After extensive wash in lysis buffer, protein samples were mixed with 5× Laemmli buffer (62.5% glycerol, 12.5% SDS, 0.5% bromophenol blue, 25% fresh 2-mercaptoenthanol in 30 mM Tris-HCl, pH 6.8) at room temperature for 5 min, and separated from beads using microcentrifuge spin columns (Thermo Scientific).

**Immunoblotting**. Protein samples were separated by SDS-PAGE, transferred onto 0.45 μm nitrocellulose membranes (PerkinElmer) and blocked with 5% non-fat dry milk dissolved in a Tris-based buffer (10 mM Tris, 150 mM NaCl, and 0.1% Tween 20, pH 7.4) at room temperature for 1 h. Human PANX1 proteins were detected by incubating with anti-Panx1 antibody (Cell Signaling Biotechnology; #91137; 1:1000). Anti-FLAG M2 (Sigma–Aldrich, #F3156; 1:1000) was used to detect Myc- and FLAG-tagged α1D-AR proteins or FLAG-tagged HDAC6. As a loading control for cell-surface biotinylation samples, we used an antibody to Na⁺/K⁺-ATPase (Cell Signaling, #3010; 1:1000). Anti-acetylated lysine antibody (Cell Signaling, #9441; 1:1000) was used to probe lysine acetylation on hPANX1 proteins. Anti-β-actin-peroxidase antibody (Sigma–Aldrich; #A3854; 1:50,000) was used as a loading control. Amersham ECL horseradish peroxidase (HRP)-linked secondary antibodies (GE Healthcare; anti-Rabbit IgG: NA9340V or anti-Mouse IgG: NA931V; 1:6000–10,000) and Western Lightning Plus ECL were used to visualize immunoreactive signals on Amersham Hyperfilm ECL (GE Healthcare). Acetylated lysine signal was imaged by using ChemiDoc™ Imaging Systems (BioRad). Images of co-immunoprecipitated HDAC6-FLAG and PANX1-Strep were taken by using ImageQuant LAS4000 Mini (GE Healthcare). Quantification of immunoreactive signal was performed by using ImageJ software (v1.8; NIH).

**Measurement of intracellular release of calcium**. Gαq/11/12/13-deleted HEK293 cells[70] were transiently transfected with α1D-AR and Gαq constructs (Gαqwt or GαqA253K) using Lipofectamine 2000 as described above. Forty-eight hours after transfection, cells were trypsinized, centrifuged and resuspended in Ringer solution (155 mM NaCl, 4.5 mM KCl, 1 mM MgCl₂, 5 mM HEPES, 10 mM D-glucose, 2 mM CaCl₂) containing 5 μM Fura-2AM, 500 μM probenecid, 0.02% Pluronic F-127 (Invitrogen, P3000MP) covered from light. After 30 min incubation at room temperature, Fura-2AM-loaded cells were resuspended in DMEM containing 10% FBS, 500 μM probenecid, and $1.5 \times 10^5$ cells were plated in each well of 96-well plates (Cellvis, P96-1-N; poly-lysine-coated) and kept at room temperature for additional 20 min covered from light. FlexStation 3 (Molecular Devices) was used to detect fluorescence intensity using Softmax Pro 7 software (excitation wavelengths: 340 nm/380 nm; emission: 510 nm). Fura-2AM-loaded cells were first incubated in calcium-free Hank's buffered salt solution (155 mM NaCl, 4.5 mM KCl, 1 mM MgCl₂, 5 mM HEPES, 10 mM D-glucose, 0.5 mM EDTA) for 2 min to obtain baseline fluorescence intensity, followed by application of 20 μM PE or 1 μM thapsigargin. The peak intensity of PE- or thapsigargin-triggered release of intracellular calcium was determined by averaging at least three independent measurements. Subsequently, extracellular free calcium (2 mM) and ionomycin (5 μM) were applied sequentially (as controls) to determine the amplitude of store-operated $Ca^{2+}$ entry and the maximum intensity of $Ca^{2+}$-dependent fluorescence in the cells.

**Stable isotope labeling with amino acids in cell culture**. HEK293T cells were cultured in DMEM containing 10% FBS, with either L-lysine-2HCl/L-arginine-HCl (light labeled) or $^{13}C_6$ $^{15}N_2$ L-lysine-2HCl/$^{13}C_6$ $^{15}N_4$ L-arginine HCl (heavy labeled) (Thermo Fisher Scientific, A33972) for >14 days at 37 °C with humidified air containing 5% $CO_2$. To identify potential lysine acetylation residues of PANX1, differentially labeled cells were separately transfected with α1D-AR and Strep-tag-conjugated hPANX1 using Transporter™ 5 (Polysciences). Twenty-four hours after transfection, both groups of cells were washed once with HEPES-based bath solution (above) and then incubated in that solution for additional 15 min at room temperature. Heavy-labeled cells were then further incubated in bath solution containing 20 μM PE, whereas light-labeled cells were separately incubated in bath solution alone, both at room temperature for 10 min.

Two groups (heavy labeled and light labeled) of cells were independently lysed using a lysis buffer composed of 150 mM NaCl, 50 mM Tris-HCl, 0.5% CHAPS, 5 mM EDTA, Halt™ protease and phosphatase inhibitor cocktail (Thermo Scientific; #78440), 5 μM trichostatin A, and 5 μM tubacin. Equal amounts of two protein samples were pooled and incubated with Strep-Tactin Superflow Plus beads

(Qiagen; #30002) at 4 °C with constant rocking for >2 h. Precipitated protein samples were washed twice using Wash Buffer A (250 mM NaCl, 50 mM Tris-HCl, and 0.1% CHAPS, pH 7.4), followed by two washes with Wash Buffer B (150 mM NaCl, 50 mM Tris-HCl and 0.1% CHAPS, pH 7.4). Protein samples were reconstituted in GlycoBuffer 2 (New England BioLabs), and mixed with 5 μM trichostatin A, 5 μM tubacin, and Halt™ protease and phosphatase inhibitor cocktail along with PNGase F (New England Biolabs; #P0708; 40 Unit μL−1) to remove glycosylation of hPANX1 proteins. After >2 h incubation at room temperature, samples were washed twice using Wash Buffer C (500 mM NaCl, 50 mM Tris-HCl, and 0.4% CHAPS, pH 7.4), followed by two further washes with Wash Buffer B. All wash steps were done at 4 °C with constant rocking.

To collect hPANX1 proteins with or without PE stimulation, precipitated proteins were eluted from Strep-Tactin beads by using Wash Buffer A containing 10 mM desthiobiotin, collected by using microcentrifuge spin columns, and concentrated by using Amicon Ultra 0.5 mL filter unit (Millipore Sigma; 30KDa cutoff). Concentrated protein samples were boiled in 6× Laemmli buffer for >5 min, followed by SDS-PAGE separation using Mini-PROTEAN TGX™ precast gels (BioRad; 10%). Colloidal blue staining (Invitrogen; #LC6025) was performed to visualize precipitated protein samples prior to mass spectrometry analysis.

**Mass spectrometry**. Colloidal blue-stained gel bands were destained in 25 mM $NH_4HCO_3$, 50% acetonitrile, pH 8.3. After vacuum drying, the gel bands were digested with 100 μL of trypsin/Lys-C (Promega) (20 μg mL$^{-1}$ in 25 mM $NH_4HCO_3$) at 37 °C, overnight. The digested peptides were extracted with 100 μL 70% acetonitrile, 5% formic acid. Each peptide sample was desalted (ZipTips, EMD Millipore), dried by vacuum centrifugation, and resuspended in 16 μL 0.1% TFA. Samples were loaded on a C18 nanobore trap column (Acclaim PepMap100 C18, 2 cm, nanoViper, Thermo Scientific), and resolved on a C18 Easy-Spray column (Acclaim PepMap RSLC C18, 2 μm, 100 Å, 75 μm × 500 mm, nanoViper, Thermo Scientific) by nanoflow LC (EASY-nLC 1200, Thermo Fisher Scientific) coupled online with an Orbitrap Fusion Lumos Tribrid MS (Thermo Fisher Scientific). The peptides were resolved using a linear gradient of 2% mobile phase B (95% acetonitrile with 0.1% formic acid) to 32% mobile phase B within 60 min at a constant flow rate of 250 nL min$^{-1}$. The C18 Easy-Spray column was heated at 50 °C during the analysis. The 12 most intense molecular ions in each MS scan were sequentially selected for high-energy collisional dissociation (HCD) using a normalized collision energy of 35%. The mass spectra were acquired at the mass range of $m/z$ 400–1600. The Easy-Spray Ion Source (Thermo Scientific) capillary voltage and temperature were set at 2.0 kV and 275 °C, respectively. Dynamic exclusion (15 s) was enabled to minimize redundant peptide fragmentation events. The RF lens was set to 30% during the MS analysis and both MS1 and MS2 spectra were collected in profile mode. Data were searched against a Swiss-Prot human protein database (http://www.uniprot.org/uniprot/) using Proteome Discoverer (v.2.1.1.21, Thermo Fisher Scientific) via Mascot (v. 2.5.1, Matrix Science Inc.) with the automatic decoy search option set followed by false-discovery rate (FDR) processing by Percolator (v.3.5). The exact Panx1 (Q96RD7) protein sequence used in the experiments (including the engineered tag and spacer) was added to the Swiss-Prot human protein database. Data were searched with a precursor mass tolerance of 10 ppm and a fragment ion tolerance of 0.05 Da, a maximum of two tryptic mis-cleavages and dynamic modifications for oxidation (15.9949 Da) on methionine residues and for acetylation (42.01056 Da) on lysine residues. The heavy SILAC labeled lysine (8.014199 Da) and arginine (10.008269 Da) residues were also set for dynamic modification. Resulting peptide spectral matches (PSMs) were filtered using an FDR of ≤1% (Percolator q-value ≤ 0.01). Although trypsin may cleave less efficiently at acetylated lysine residues, we included MS identification of peptides with C-terminally acetylated lysines (i.e., at the trypsin/Lys-C cleavage site) since the modified Lys residues were clearly defined by their spectral characteristics and were observed in all three analyses. Spectrum profile is available at Proteomics Identification Database (PRIDE; Project accession: PXD025912).

**Statistics**. Data are presented as mean ± s.e.m. Statistical analyses, described in figure legends, were performed using GraphPad Prism (v.9).

**Reporting summary**. Further information on research design is available in the Nature Research Reporting Summary linked to this article.

## Data availability
All data generated in this study are provided in the Supplementary Information/Source Data file. The raw mass spectrometry proteomics data have been deposited to the ProteomeXchange Consortium via the PRIDE[71] partner repository with the dataset identifier PXD025912. Source data are provided with this paper.

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

## Acknowledgements

This work was supported by P01 HL120840 (to U.M.L., B.N.D., K.S.R., and D.A.B.); by a Pharmacological Sciences Training Grant T32 GM007055 (to E.C.G., J.K.S., C.B.M., and C.A.D.); by the Ministry of Science and Technology, Taiwan (MOST 108-2320-B-007 -007 -MY2, to Y.H.C.); by the National Tsing Hua University, Taiwan, through the Ministry of Education's Higher Education Sprout Project (110Q2705E1 to Y.H.C.); and by predoctoral fellowships from the American Heart Association (PRE223005, to J.K.S.) and the National Cancer Institute (CA236370, to A.K.N.). We thank members of the University of Virginia Pannexin Interest Group for helpful comments throughout the course of the study. We thank Dr. Patricia Hinkle (University of Rochester, Rochester, NY) and Dr. Asuka Inoue (Tohoku University, Miyagi, Japan) for generously providing cell lines.

## Author contributions

Y.-H.C and D.A.B. developed concept and designed experiments. Y.-H.C., J.K.S., and A.K.N. performed electrophysiological experiments. Y.-H.C., E.C.G., H-Y.G., and S.Y.G. carried out cell-surface biotinylation and western blotting. C.A.D. performed Ca²⁺ assay, C.B.M. isolated mouse T lymphocyte and generated PANX1-deleted Jurkat cells using CRISPR-Cas9 technique. M.K. generated Jurkat cell line stably expressing Cas9 proteins. Y.-H.C. and H-Y.G. purified PANX1 proteins, and M.Z. and T.P.C. carried out mass spectrometry experiments and the associated data analysis. Y.-H.C., T.P.C., U.M.L., B.N.D., K.S.R., and D.A.B. supervised the experiments. Y.-H.C. and D.A.B conceptualized and wrote the manuscript, and all authors edited the manuscript.

## Competing interests

The authors declare no competing interests.
