## [Peer Review File · Nature Communications]

Reviewers' comments:

Reviewer #1 (Remarks to the Author):

Major concerns:

Physiological role is not clear. Why compare current densities at +80mV, which is clearly not physiological? Are changes in current density significant at potentials closer to the resting membrane potential or in the normal range that non-excitable cells such as T lymphocytes would see? What is the role of the open Panx1 channels in this context? Are they releasing ATP as implied or is there another important physiological response? This should be evaluated in T lymphocytes.

The data as presented do not rule out two alternative possibilities (which are not mutually exclusive and may be acting in tandem). First, that lysine-deacetylation is a regulator of a more conventional posttranslational modification of pannexin, such as phosphorylation, that induces channel opening. Second, that the RhoA pathway leading to deacetylation is also directly stimulating a kinase pathway. These two possible alternative explanations must be addressed:

There are numerous reports showing that RhoA couples to Src kinase through mDia (for example Veetill et al., J Virol 2006), leaving open the possibility that a parallel mDia mechanism is at play and that HDAC6 mediated deacetylation is regulating either access of kinases such as Src to Panx1 or mDia is directly activating Src, leading to phosphorylation of Panx1. The combination of these 2 events is also possible. This latter point is supported by the data presented in Supp Fig 4 where there is $\alpha 1$ -ADR activation of Panx1 in the K140Q and K140/409Q mutations. This point is glossed over on pg. 6 where the authors state: "In addition, they also imply that alternative, non-HDAC6 mechanisms may contribute to receptor activation of these Gln-substituted channels."

It is surprising that effects of Src are not considered in the manuscript because of publications showing that Src is important for regulating the activity of pannexin, and two Src sites at Y198 (DeLallo et al JBC 2019) and Y308 (Weilinger et al Nat Neuro 2016). Experiments should be included to show that $\alpha 1$ D-AR is / is not affecting Src activation and if it is, the dependence on RhoA, mDia and HDAC6 needs to be evaluated. The phosphorylation state of Panx1 needs to be evaluated, including in the Lys-140 and Lys-409 (both R and Q) mutant channels. Further, the effect of inhibiting Src activity (pharmacologically or with dominant negative constructs) needs to be included. Although some papers have suggested Src is not involved in $\alpha 1$ -AR activation of pannexin, there is clear indication throughout the pannexin literature that regulation of the pannexin channel is occurring in a cell type specific way and the authors must rule out the possibility that alterations of Ac-Lys is not leading to activation by Src and that mDia is not directly activating Src in parallel to HDAC6.

The main conclusion of the paper is that a novel, reversible activation mechanism for Panx1 (and possibly other channels) has been identified. I see no data showing reversibility of the $\alpha 1$ -ADR deacetylation of pannexin. For example, in Figure 6 does the CBX-sensitive current in CD4+ T cells reverse upon removal of PE? In general, for deacetylation to be considered a novel and important mechanism for regulating channels the authors need to identify the acetylase and ideally, a mechanism for its activation and silencing of the channel. Without these data the impact of the findings in a physiological context are greatly diminished.

Specific concerns:

Figure 1: in the example shown in a, the NPo looks to be much higher than the mean of ~ 0.15 and 0.3 during PE that is shown in b. Perhaps more representative traces could be selected. There also seems to be a disconnect between the NPo and the whole cell recordings in that there is substantial basal activity of mPanx1 but NPo under basal and CBX conditions is similar. I would have expected the basal to be intermediate between the +PE and +CBX conditions as the whole-cell I-V curves show. What is the source of this discrepancy? In 1g, the time course of inhibition looks very different than the other plots in the figure, what is the peak in current that appears after CBX is applied? Fig 1h provides the initial rationale for looking at a Rho pathway and not a PLC pathway. However, there is

no positive control to show that the GqA253K mutant is behaving as expected. Evidence showing a lack of downstream Gq-mediated signalling (other than Panx1) and a persistent PLC-mediated Ca²⁺ response should be provided.

Figure 2: It is surprising that apparently all cells showed effects of C3 exoenzyme and RhoA(T19N) coexpression. This is because triple transfections have a low probability of getting all 3 proteins expressed in a single cell. Was there a transfection marker for each of the proteins? How often was no effect (i.e. lack of CBX sensitive current upon PE administration) seen? It seems that cells that did respond to PE, as would be expected if C3 exoenzyme / RhoA(T19N) were not expressing, are excluded from analysis. How was it determined in these cells that a failure to inhibit was due to lack of expression and not lack of efficacy? Figure 1f should be supported by controls to show that RhoA(G14V/F39A) is not activating other RhoA targets than mDia.

Minor:

The assumption in para 1 of the Results that the channel and α -1DR are physically separated and thus the message is diffusible is not supported by data. PANX1 is part of several protein complexes (i.e. with P2XR and NMDARs). It is conceivable that α -1DR and PANX1 are complexed together.

Top of page 4: "...that is independent of PLC-Ca²⁺ and likely involves activation of RhoA." The statement 'likely involves RhoA' is premature at this point in the manuscript. Ruling out Ca²⁺ and PLC does not automatically rule in RhoA.

Pg 4, 2nd paragraph: "These data indicate that RhoA activation is necessary and sufficient for receptor-mediated PANX1 activation." It would be more precise to replace receptor-activated with Gq-activated or α -1ADR-activated to avoid the impression that all receptors use this mechanism.

Pg 4, end of "RhoA is required section" The authors state "Thus, we found large PANX1 currents under unstimulated conditions in cells transfected with RhoA(G14V/F39A), and those currents were not further increased by PE stimulation." How was occlusion of the response determined? How was a ceiling effect ruled out? The second point is critical given the large amplitude of the currents in the basal condition. This query is also relevant for Fig 3a and b and the first paragraph of pg 5.

Pg 5, 2nd paragraph: "and by tubacin, a more selective HDAC6 inhibitor." What is meant by 'more selective'?

Figure 3: Why are the I-V curves in g and h shown as mean with error? This is not the trend elsewhere. Further, there are labeling inconsistencies throughout the figures – in some panels traces are labeled and others they are not, for example.

In Fig 3d and on pg 5 the authors state that Panx1 in patches had no activity as previously reported. However in the methods on pg 13 (last paragraph) it is stated that only patches without initial channel activity were selected. Where there instances when excised patches had channel activity and how was it evaluated if these were Panx1 channels or not?

The blot in Fig 5b is underwhelming and the changes in Ac-Lys (specifically in the double mutant) are not convincing and the intensity of the AC-Lys bands appears barely above background. How were the ROIs for the bands determined with such 'fuzzy' edges? How was quantification in the right panel determined? The y-axis states it is a ratio / percentage but I find it very hard to believe that the Ac-Lys signal is 80% of the FLAG signal.

Lys-409 is not conserved in the mouse. Is Lys-140? This will speak to activation mechanisms in the mouse versus human channels.

Pg 7, T-lymphocyte section, first sentence. What is meant by relatively high levels of pannexin?

What temperature where the whole-cell recordings performed at?

Are there other CBX-sensitive channels in the cells evaluated?

Experiments in Supp Fig 1d and e are not well described in the text and need to be placed into context.

Reviewer #2 (Remarks to the Author):

The manuscript by Yu-Hsin Chiu and the colleagues studied the activation mechanism of ion channel PANX1 by alpha1D-AR. To demonstrate that RhoA is both necessary and sufficient for the activation process, the authors applied the overexpression of wildtype and dominant negative mutant of RhoA as well as the overexpression of C3 exoenzyme that inactivated RhoA through ADP ribosylation. They further showed that one of the downstream effectors of RhoA, mDia, and a lysine deacetylase HDAC6, are involved in the activation of PANX1. To demonstrate that HDAC6-mediated deacetylation is involved in the activation process, the authors showed that in vitro HDAC6 incubation with inside-out membrane promoted the activation of the channel. Using SILAC and MS analysis, the authors identified K140 and K409 acetylation levels were reduced upon channel activation. Mutations of both sites to Arg resulted in a constitutively activated PANX1 while mutations to Gln abolished the increase in the channel activity mediated by HDAC6. The authors further studied the activation of PANX1 in T lymphocytes cells.

The data quality of the overall study was high and the integrated analysis revealed a potentially novel mechanism involving HDAC6 and deacetylation in the activation of PANX1. However, additional mechanistic studies should be performed to convincingly demonstrate the mechanism in this process. The major and minor concerns are listed below:

Major concerns:

1. Based on the data presented in the manuscript, acetylation of PANX1 appears to block its basal activation and removal of acetylation by HDAC6 activates the channel. If this is true, the basal level of acetylation on the expressed PANX1 should be very high or nearly 100% and HDAC6 should result in an apparent decrease in acetylation level on PANX1. Could these be tested and confirmed? In addition, it is important to show that the enzyme-dead mutant of HDAC6 and treatment of HDAC6 inhibitors prevents the activation of the channel.
2. SILAC-based quantification showed the decrease of acetylation on two sites on PANX1 following activation. Since these are key findings, mass spectrometry data should be included in the figure to demonstrate the change. It is also important to note that quantification of site-level changes should be normalized against the protein-level changes based on SILAC quantification since the cell numbers and expression levels could be different.
3. Additional quantitative MS experiments should be included to test the changes in acetylation on each site after HDAC6 incubation.

Minor concern:

1. It is also important to demonstrate that PANX1 can directly interact with HDAC6 if this is not yet known.

Reviewer #3 (Remarks to the Author):

Deacetylation as a receptor-regulated...

By Chiu et al

In a careful and decisive manner, the authors demonstrate that HDAC6 can activate PANX1 directly by

deacetylation of Lys140 and that reversible acetylation-deacetylation of the channels regulates the channel on heterologous expression. Further, the authors offer strong evidence for the operation of this non-canonical and previously unknown mechanism in T lymphocytes. The work is impressive as well for its clarity of presentation. After using pharmacology to hypothesize the mechanism, the authors use MS to find the sites of interest and then point mutation to identify the residue controlling the regulatory pathway under study. Particularly elegant, they first show the effects require a diffusible signal by a classical strategy using on-cell patch mode to isolate the channel from activator applied in the bath solution and then pull membrane patches off cells and apply the regulatory enzyme to the cytoplasmic face of the membrane to modulate the channels directly. I have only one minor comment: in Figure 1 the blocker carbenoxolone, CBX, is used without being discussed except in the legend; this should be enunciated since it is used throughout. Beautiful work. Well done. Steve Goldstein

We thank the Reviewers for a positive review of the manuscript, and for constructive suggestions for improvement. We have addressed all comments in detail below, and hope you agree that our responses and revisions have further improved the paper.

Here, we first note that this work identifies a novel mechanism for direct channel activation (by reversing constitutive acetylation on membrane channel lysine residues), and a novel role for HDAC6 in a non-canonical signaling pathway by Gαq-coupled receptors. We believe our data compellingly support these main conclusions of the paper. We do not, however, suggest that this is the only mechanism for GPCR-mediated Panx1 activation, and the precise role of this form of channel activation in immune cell function remain to be elucidated – based on the present work, we expect that other groups will join us in continuing to explore this and other relevant mechanisms in those contexts.

Reviewers' comments:

Reviewer #1 (Remarks to the Author):

Major concerns:

1a) Physiological role is not clear. Why compare current densities at +80mV, which is clearly not physiological? Are changes in current density significant at potentials closer to the resting membrane potential or in the normal range that non-excitabile cells such as T lymphocytes would see?

Panx1 channels are active over a wide voltage range and generate a weakly outwardly-rectifying open channel current (i.e., larger at positive membrane potentials than at negative potentials). Because of this outward rectification, there is greater signal-to-noise for currents at positive membrane potentials, like +80 mV, where we took our measures. This is not unusual – indeed, it is a generally accepted way of evaluating such rectifying currents.

However, it is also clear from the *I-V* curves presented throughout the manuscript that the channel is active at negative membrane potentials maintained by non-excitabile cells, including the particular case of T lymphocytes (as depicted in **Fig. 6**). As evident in those *I-Vs*, effects on Panx1 currents observed at +80 mV were also apparent at negative potentials. Nevertheless, we appreciate that this journal has a broad audience that may include readers who are not familiar with electrophysiological data, and may not recognize this from the *I-V* plots. Thus, we now emphasize this important point explicitly in the text (see **p.3**).

1b) What is the role of the open Panx1 channels in this context? Are they releasing ATP as implied or is there another important physiological response? This should be evaluated in T lymphocytes.

This is an important question, and one that we are actively exploring. Interestingly, despite substantial effort, we have not detected Panx1-dependent ATP release from T lymphocytes after α1-adrenoceptor stimulation, even though we can easily observe ATP release in apoptotic T cells from caspase-activated Panx1 channels and in vascular smooth muscle cell by α1-AR-activated Panx1 channels (*Nature* 467: 863-7, 2010; *Sci Signal* 8: ra17, 2015).

As discussed in the last paragraph of the manuscript (**p.10**), we are considering that channel activation in α1-AR-expressing T cells may play other roles, such as supporting acetylcholine release in cholinergic anti-inflammatory pathway, or contributing to organization of the immune synapse.

2. The data as presented do not rule out two alternative possibilities (which are not mutually exclusive and may be acting in tandem). First, that lysine-deacetylation is a regulator of a more conventional posttranslational modification of pannexin, such as phosphorylation, that induces channel opening. Second, that the RhoA pathway leading to deacetylation is also directly stimulating a kinase pathway.

We reiterate that the work presented here deals specifically with identifying a novel lysine deacetylation mechanism for channel activation. Our results demonstrating direct HDAC6-mediated Panx1 activation in excised patches, and requiring specific channel lysine residues, fully support this conclusion. Moreover, these inside-out patch experiments definitively rule out any contribution from any of these other hypothesized pathways in this case of channel activation.

In general, with this and the following comments, the reviewer hypothesizes multiple alternative pathways for channel activation downstream of GαqPCR activation. Indeed, our results support the existence of additional, possibly parallel, pathways for channel activation -- but our own ongoing work actually suggests that they may be distinct from those proposed by the Reviewer (see below). Thus, although we agree with the Reviewer that alternative pathways are likely, we also submit that elucidating those is beyond the scope of the present report. To this point, we would submit that the number of alternatives proposed by the Reviewer actually provides a compelling rationale for not trying to add such studies to the present paper, which has already identified a novel channel activation mechanism.

These two possible alternative explanations must be addressed:

There are numerous reports showing that RhoA couples to Src kinase through mDia (for example Veetill et al., J Virol 2006), leaving open the possibility that a parallel mDia mechanism is at play and that HDAC6 mediated deacetylation is regulating either access of kinases such as Src to Panx1 or mDia is directly activating Src, leading to phosphorylation of Panx1. The combination of these 2 events is also possible. This latter point is supported by the data presented in Supp Fig 4 where there is α1-ADR activation of Panx1 in the K140Q and K140/409Q mutations. This point is glossed over on pg. 6 where the authors state: "In addition, they also imply that alternative, non-HDAC6 mechanisms may contribute to receptor activation of these Gln-substituted channels."

Again, we must emphasize that our examination of this α1D-AR-initiated signaling process culminated in a definitive set of functional experiments demonstrating that purified HDAC6 is able to unsilence Panx1 channels in an excised membrane patch and in a manner dependent on specific channel lysine residues. This direct deacetylation mechanism for channel activation is unprecedented. Moreover, under these cell-free conditions, with no available ATP, any residual patch-associated Src (or any other kinase) would be unable to phosphorylate the channel.

That said, we agree that other routes from Gαq-coupled receptor to the Panx1 channel may exist in intact cells. In fact, as the Reviewer notes, our original text acknowledged the likely contributions of some other signaling pathway(s). In further deference to this concern, we have modified the final schematic to indicate the presence of additional, yet unidentified, signaling streams (see **Fig. 6g**).

It is surprising that effects of Src are not considered in the manuscript because of publications showing that Src is important for regulating the activity of pannexin, and two Src sites at Y198 (DeLallo et al JBC 2019) and Y308 (Weilinger et al Nat Neuro 2016). Experiments should be included to show that α1D-AR is / is not affecting Src activation and if it is, the dependence on RhoA, mDia and HDAC6 needs to be evaluated. The phosphorylation state of Panx1 needs to be evaluated, including in the Lys-140 and Lys-409 (both R and Q) mutant channels. Further, the effect of inhibiting Src activity (pharmacologically or with dominant negative constructs) needs to be included. Although some papers have suggested Src is not involved in α1-AR activation of pannexin, there is clear indication throughout the pannexin literature that regulation of the pannexin channel is occurring in a cell type specific way and the authors must rule out the possibility that alterations of Ac-Lys is not leading to activation by Src and that mDia is not directly activating Src in parallel to HDAC6.

The Reviewer correctly notes that the literature includes examples of Src-mediated phosphorylation of Panx1 channels; the original (and revised) manuscripts reference both papers listed by the Reviewer.

We initially considered a role for Src and the indicated Tyr-198 and Tyr-308 sites, but we no longer favor that possibility for the following reasons. First, in our MS analysis, we see no evidence for α 1D-evoked

phosphorylation of tyrosine residues on Panx1 channels (including at the conserved Tyr-199 and Tyr-309 residues of hPANX1). Second, in response to this concern, we prepared a mutated Panx1 construct that replaces the proposed Tyr-198 and Tyr-308 sites with Phe residues that cannot be phosphorylated; as depicted in Fig. R1, α 1D-induced channel activation is intact in this mPanx1(Y198,308FF) construct.

On the other hand, we do find multiple PE-stimulated Ser/Thr phosphosites on Panx1, and we are currently screening

those for possible contributions to α 1D-mediated channel activation, and testing likely kinase candidates. We believe that a suitably comprehensive examination of whether/how α 1D-induced phosphorylation could influence channel activity would be best treated in a separate, dedicated report.

3. The main conclusion of the paper is that a novel, reversible activation mechanism for Panx1 (and possibly other channels) has been identified. I see no data showing reversibility of the α 1-ADR deacetylation of pannexin. For example, in Figure 6 does the CBX-sensitive current in CD4+ T cells reverse upon removal of PE?

Our use of the term “reversible” in this context is by way of contrast with an irreversible C-terminal cleavage-based mechanism for Panx1 activation, and in consideration of the well-known reversible nature of this form of posttranslational modification (acetylation-deacetylation). With respect to HDAC6-activated PANX1, deacetylation and unsilencing of the channel in inside-out patches actually represents a reversal of the basal acetylation state imposed by the cell.

With respect to α 1-activated whole cell currents, we chose a standard experimental protocol to quantify the CBX-sensitive component of current in each cell (i.e., the α 1-activated Panx1 current), and this entailed applying the channel blocker as the current reached its peak. As noted by the Reviewer, this did not establish reversibility of the receptor-activated current. Thus, we have performed new experiments to demonstrate reversibility of the α 1-activated current. In Fig. S1c, we now show that the PE-activated whole cell current returned toward baseline after washing the agonist (by ~56% in ~7 min, n=9).

In general, for deacetylation to be considered a novel and important mechanism for regulating channels the authors need to identify the acetylase and ideally, a mechanism for its activation and silencing of the channel. Without these data the impact of the findings in a physiological context are greatly diminished.

We strongly disagree with this general assertion. If that were true, then all kinase-mediated, phosphorylation-based channel activation mechanisms would be considered suspect before simultaneous identification of a corresponding phosphatase. This is clearly not the case. As one relevant example, the Src-mediated Panx1 activation mechanism referenced earlier by the Reviewer is likely considered impactful, even as no associated tyrosine phosphatase was identified in that study. Other examples abound. Also, from a practical standpoint, establishing the specific acetylase is not a trivial undertaking -- there are many candidates, and few specific reagents (e.g., see *Nature Communications* 8: 1527, 2017).

Specific concerns:

1. *Figure 1: in the example shown in a, the NPo looks to be much higher than the mean of ~0.15 and 0.3 during PE that is shown in b. Perhaps more representative traces could be selected. There also seems to be a disconnect between the NPo and the whole cell recordings in that there is substantial basal activity of mPanx1 but NPo under basal and CBX conditions is similar. I would have expected the basal to be intermediate between the +PE and +CBX conditions as the whole-cell I-V curves show. What is the source of this discrepancy?*

It is hard to ensure that the stochastic activity of channels in each patch will mimic perfectly the ensemble behavior of hundreds of channels in whole cell recordings. Please note that the exemplar channel recording in **Fig. 1a** matches precisely the Reviewer's expectations, and is consistent with the whole cell data. We have now marked the NPo plot to indicate the data points derived from the example.

Meanwhile, we thank the Reviewer for drawing our attention to the summary data in **Fig.1b**. We noticed that that the y-axis of **Fig. 1b** was originally mislabeled as NPo, and has now been corrected to maximum Po (derived from NPo divided by the maximum numbers of channel observed in a given patch).

2. *In 1g, the time course of inhibition looks very different than the other plots in the figure, what is the peak in current that appears after CBX is applied?*

This is a transient instability, which is not uncommon in whole cell recordings.

3. *Fig 1h provides the initial rationale for looking at a Rho pathway and not a PLC pathway. However, there is no positive control to show that the GqA253K mutant is behaving as expected. Evidence showing a lack of downstream Gq-mediated signalling (other than Panx1) and a persistent PLC-mediated Ca²⁺ response should be provided.*

Please note that we provide the requested positive control for GqA253K in Fig1j,k, which shows that this point-mutated Gαq subunit can indeed rescue PLC-mediated Ca²⁺ response in Gαq-deleted cells.

4. *Figure 2: It is surprising that apparently all cells showed effects of C3 exoenzyme and RhoA(T19N) co-expression. This is because triple transfections have a low probability of getting all 3 proteins expressed in a single cell. Was there a transfection marker for each of the proteins? How often was no effect (i.e. lack of CBX sensitive current upon PE administration) seen? It seems that cells that did respond to PE, as would be expected if C3 exoenzyme / RhoA(T19N) were not expressing, are excluded from analysis. How was it determined in these cells that a failure to inhibit was due to lack of expression and not lack of efficacy?*

We did not exclude any cells from the analysis. The recordings from control cells were interleaved with those from C3- or RhoA(T19N)-transfected cells, performed on the same day. The frequency and magnitude of responses for α1-activation of Panx1 can be gleaned from the data points presented within the first histogram of **Fig. 2c**.

We did not have a transfection marker for each protein. However, transfection efficiency of 293T cells is very high and, in general, cells that take up any DNA also take up other plasmids in the transfection admixture. So, the probability of transfected cells expressing all three plasmids is usually very high. Moreover, the cDNA ratios in the transfection mixture (with equal total cDNA) were established such that the channel was provided at the lowest concentration, then the modulator and the receptor at the highest concentration. Since these experiments were performed with mPanx1 that generates basal CBX-sensitive current, we could directly establish that the channel is present (with the lowest amount of cDNA) and ensure greater probability that the higher concentration cDNAs were also transfected. We now detail this transfection strategy more clearly in the Methods (see **p.14**).

5. Figure 1f (2f) should be supported by controls to show that RhoA(G14V/F39A) is not activating other RhoA targets than mDia.

The RhoA(G14V/F39A) mutation has been extensively characterized (see *J Biol Chem*, 278: 49911, 2003; *Nat Cell Bio*, 3: 723, 2001). It is inactive against a host of RhoA effectors (i.e., ROCK, RhoGDI, Citron, PKN, Kinectin) while maintaining its activation of mDia. It is not clear which other RhoA effector should be tested, or how that would rule out actions on any additional RhoA effectors that have yet to be discovered. Most importantly, this result was taken only to suggest that mDia may be involved (see **p.5**), a possibility tested more directly in subsequent experiments.

Minor:

1. The assumption in para 1 of the Results that the channel and α -1DR are physically separated and thus the message is diffusible is not supported by data. PANX1 is part of several protein complexes (i.e. with P2XR and NMDARs). It is conceivable that α -1DR and PANX1 are complexed together.

We do not make this assumption or this claim. Indeed, we agree that these results cannot rule out the possible existence of an α 1DR-channel complex.

Rather, these data demonstrate that channel activation measured under these cell-attached conditions, when the liganded α 1D receptor is physically separated from the recorded channel in the patch pipette, must involve a diffusible messenger. This is a classical approach for demonstrating involvement of a diffusible messenger (see *Proc Biol Sci* 250: 119-125, 1992; also see similar schematic in *Nature Rev Neurosci* 6: 850–862, 2005).

For clarity, we have re-worded the description of the schematic in the figure legend (**Fig. 1**, see **p.25**).

2. Top of page 4: "...that is independent of PLC-Ca²⁺ and likely involves activation of RhoA." The statement 'likely involves RhoA is premature at this point in the manuscript. Ruling out Ca²⁺ and PLC does not automatically rule in RhoA.

We have re-worded to emphasize that it is specifically the results from the GqA253K re-expression experiment that point to RhoA as the potential mediator – a possibility tested more directly in subsequent experiments (**p.3**). We also substituted "perhaps" for likely in the offending clause.

3. Pg 4, 2nd paragraph: "These data indicate that RhoA activation is necessary and sufficient for receptor-mediated PANX1 activation." It would be more precise to replace receptor-activated with Gq-activated or α -1ADR-activated to avoid the impression that all receptors use this mechanism.

This change in wording has been made.

4. Pg 4, end of "RhoA is required section" The authors state "Thus, we found large PANX1 currents under unstimulated conditions in cells transfected with RhoA(G14V/F39A), and those currents were not further increased by PE stimulation." How was occlusion of the response determined? How was a ceiling effect

ruled out? The second point is critical given the large amplitude of the currents in the basal condition. This query is also relevant for Fig 3a and b and the first paragraph of pg 5.

Indeed, it can be difficult to discriminate an occlusive situation (i.e., channel activated by the same mechanism, so no further effect possible) from a ceiling effect (i.e., basal channel activity pegged at some high level due to effects of an independent mechanism.)

In this case, our data support a common mechanism – i.e., occlusion. The critical observations are presented in **Fig. 3g-i**. In these experiments, the high basal channel activity typically associated with either RhoA(G14V/F39A) or with DAD was reduced by tubacin. Under these conditions, PE had essentially no effect on Panx1 channels even though the basal currents were a much lower starting point (i.e., not at the ceiling). We now provide a more extended description of this interpretation (see **p.5**).

5. Pg 5, 2nd paragraph: “and by tubacin, a more selective HDAC6 inhibitor.” What is meant by ‘more selective’?

We had intended for this to refer to the immediately preceding description of the effects of trichostatin A, which we note is a broad spectrum HDAC inhibitor. This has now been clarified (**p.5**).

6. Figure 3: Why are the I-V curves in g and h shown as mean with error? This is not the trend elsewhere. Further, there are labeling inconsistencies throughout the figures – in some panels traces are labeled and others they are not, for example.

For most of the data presentation, the *I-V* curves show effects of drug treatments within the same cell (i.e., PE, CBX), and grouped data are provided in corresponding histogram plots. The *I-V* data presented in **Fig. 3g,h** compare across experimental groups (i.e., not within a cell). Thus, this format allowed us to compare averaged currents from all RhoA(G14V/F39A)-expressing and DAD-expressing cells, either with or without tubacin, across the whole voltage range and without duplicating histograms from the groups without tubacin (i.e., from **Fig. 2g, Fig. 3c**).

Together with the consistent color coding of the traces, we have now added labels to every trace.

7. In Fig 3d and on pg 5 the authors state that Panx1 in patches had no activity as previously reported. However, in the methods on pg 13 (last paragraph) it is stated that only patches without initial channel activity were selected. Where there instances when excised patches had channel activity and how was it evaluated if these were Panx1 channels or not?

We consistently find no evidence for basal hPANX1 currents in HEK293T or Jurkat cells, or for basal mPanx1 currents in primary mouse T cells, as determined based on sensitivity to CBX. Thus, for these experiments, any initial channel activity is unlikely to be due to Panx1 channels. In addition, excluded channels had obvious differences in channel amplitude (unitary conductance) from Panx1 channels activated by α 1D-AR or by caspase/TEVp (see **Figs. 4-6**; also *Nature Communications* 8: 14324, 2017). We now provide this information in the Methods (**p.14**).

8. The blot in Fig 5b is underwhelming and the changes in Ac-lys (specifically in the double mutant) are not convincing and the intensity of the AC-Lys bands appears barely above background. How where the ROIs for the bands determined with such ‘fuzzy’ edges? How was quantification in the right panel determined? The y-axis states it is a ratio / percentage but I find it very hard to believe that the Ac-Lys signal is 80% of the FLAG signal.

The normalization was first relative to FLAG (within an experiment), and then to the peak intensity (for comparison across experiments, i.e., biological replicates). We now provide this normalization procedure in the figure legend (**Fig. 5b, p.26**).

The quantification was performed in ImageJ, as now outlined in the revised Methods (p.16). In addition, please see **Fig. R2** for an example of the quantification procedure that illustrates how ROI borders were determined, and how the associated optical densities were calculated.

Figure R2. Quantification of acetylated-lysine blots. To quantify immunoreactivity in Western blots by ImageJ (v.1.8), we took the area under curve (AUC) to define the signal intensity in areas enclosed by blue or pink boxes. Signal intensity obtained from the mock transfection control (first box from left) was taken as background for individual blots. Background-corrected signal of AUC1 (i.e., BC; signal from acetylated proteins) was first normalized to the corresponding AUC2 (signal of FLAG-tagged PANX1), and then normalized to the highest signal ratio (BS/AUC2) within a given experiment.

9. *Lys-409 is not conserved in the mouse. Is Lys-140? This will speak to activation mechanisms in the mouse versus human channels.*

Yes, Lys140 is present in the mouse sequence – this is now stated explicitly (p.6).

10. *Pg 7, T-lymphocyte section, first sentence. What is meant by relatively high levels of pannexin?*

We now provide a reference for stating that Panx1 and $\alpha 1$ receptors are expressed in T cells; we no longer refer to high levels of Panx1 in those cells (although we note here that there is a high level of Panx1-dependent dye uptake in apoptotic T cells).

11. *What temperature where the whole-cell recordings performed at?*

All recordings were performed at room temperature, as now indicated in the Methods (p.13).

12. *Are there other CBX-sensitive channels in the cells evaluated?*

The $\alpha 1$ -activated current is completely blocked by CBX in primary T lymphocytes and in Jurkat cells.

To obtain further evidence that the $\alpha 1$ -activated current in T cells is due to PANX1, we generated a new stable Jurkat cell line in which the channel was eliminated by CRISPR/Cas9-mediated deletion. As depicted in a new supplementary figure (Suppl. Fig. 6), PE-activated current was not observed in Jurkat cells after CRISPR deletion of Panx1.

13. *Experiments in Supp Fig 1d and e are not well described in the text and need to be placed into context.*

We have now revised to provide more rationale for showing that mGluR and H1 receptors can activate Panx1 channels (see p.3).

Reviewer #2 (Remarks to the Author):

The manuscript by Yu-Hsin Chiu and the colleagues studied the activation mechanism of ion channel PANX1 by $\alpha 1$ -AR. To demonstrate that RhoA is both necessary and sufficient for the activation process, the authors applied the overexpression of wildtype and dominant negative mutant of RhoA as well as the overexpression of C3 exoenzyme that inactivated RhoA through ADP ribosylation. They further

showed that one of the downstream effectors of RhoA, mDia, and a lysine deacetylase HDAC6, are involved in the activation of PANX1. To demonstrate that HDAC6-mediated deacetylation is involved in the activation process, the authors showed that in vitro HDAC6 incubation with inside-out membrane promoted the activation of the channel. Using SILAC and MS analysis, the authors identified K140 and K409 acetylation levels were reduced upon channel activation. Mutations of both sites to Arg resulted in a constitutively activated PANX1 while mutations to Gln abolished the increase in the channel activity mediated by HDAC6. The authors further studied the activation of PANX1 in T lymphocytes cells. The data quality of the overall study was high and the integrated analysis revealed a potentially novel mechanism involving HDAC6 and deacetylation in the activation of PANX1. However, additional mechanistic studies should be performed to convincingly demonstrate the mechanism in this process. The major and minor concerns are listed below:

Major concerns:

1. Based on the data presented in the manuscript, acetylation of PANX1 appears to block its basal activation and removal of acetylation by HDAC6 activates the channel. If this is true, the basal level of acetylation on the expressed PANX1 should be very high or nearly 100% and HDAC6 should result in an apparent decrease in acetylation level on PANX1. Could these be tested and confirmed?

The Reviewer is correct in suggesting that these results predict a very high fraction of PANX1 channels in a basally acetylated state – particularly if one assumes that this is the only channel silencing mechanism. However, this may not be the case.

Indeed, other channel activating mechanisms likely exist and those may rely on distinct silencing mechanisms (Reviewer 1, Point #2); de-acetylation would be effective only on the subset of channels silenced by acetylation. Please note that our functional studies do not relate the de-acetylated channel current to some independent measure of total channel current (e.g., as measured by another activation mechanism), and thus an unknown fraction of channels may be resistant to activation by HDAC6.

More generally, it is not certain how the degree of functionally-relevant acetylation on cell-membrane localized Panx1 can be established in a strict quantitative sense. For the MS analysis, it is not possible to compare relative fraction of acetylated vs non-acetylated peptides since they do not have the same ionization potential. In addition, the basal level of acetylation will include other acetylated lysine residues that appear to be dispensable for channel silencing (i.e., those other than Lys-140, and perhaps Lys-409). A further complication is that the subunit stoichiometry of acetyl-lysine moieties required for silencing has not been established. Thus, a strict channel-level (or even subunit-level) stoichiometric quantification may prove difficult to achieve, and then to interpret.

Nonetheless, it is clear that some level of constitutive acetylation of Panx1 channels would be expected on Lys-140 and Lys-409 of PANX1, as demonstrated in the experiments of **Fig. 5b**.

In addition, it is important to show that the enzyme-dead mutant of HDAC6 and treatment of HDAC6 inhibitors prevents the activation of the channel.

We showed that expression of the catalytically-dead HDAC6 has no effect on whole cell PANX1 currents, and that expression of the active HDAC6 does not activate whole cell currents from the KKQQ mutated PANX1 channels (**Fig. 5f-h**).

For the excised patch experiments, we do not have access to a commercial supply of the purified, catalytically-dead HDAC6. For this reason, we chose instead to test the KKQQ mutant channels. This had the advantage of simultaneously implicating the enzymatic activity of the HDAC6, while also testing deacetylation of the channel itself (**Fig. 5j**).

2. SILAC-based quantification showed the decrease of acetylation on two sites on PANX1 following activation. Since these are key findings, mass spectrometry data should be included in the figure to demonstrate the change. It is also important to note that quantification of site-level changes should be normalized against the protein-level changes based on SILAC quantification since the cell numbers and expression levels could be different.

We now provide the abundance changes determined for the K140- and K409-acetylated versions of the cognate peptides (legend to **Fig. 5a**), normalized to the median abundance of PANX1 and as a ratio of PE-treated:vehicle-treated (K140-Ac: 0.875; K409-Ac: 0.912).

With respect to these abundance changes, some caveats must be acknowledged. First, the immunoprecipitation did not discriminate between membrane-associated and other PANX1 channels, whereas the functional effects we measured would only involve the plasma membrane-associated channel fraction. If channel acetylation takes place at the cell surface, the “dilution” of the relevant signal could be >30-fold, since we find the Panx1 immunoreactive signal from cell membranes (1 mg loaded) is always lower than that from the cell lysate (30 µg loaded) in our cell surface biotinylation experiments. Also, as alluded to above, the subunit stoichiometry for this modification, and for relief of acetylation-based silencing, is not known. This might be amenable to experiments using concatenated channels with varying numbers of modified subunits (an approach we used previously in the context of PANX1 C-tail cleavage; see *Nature Communications* 8: 14324, 2017). Regardless, if we take these data at face value, assuming these are hexameric channels with equal subcellular access to the modifying enzymes in the presence and absence of PE, it might suggest that removal of a single acetyl group from Lys-140 was sufficient for channel unsilencing (i.e., leaving 5/6 K140-Ac intact, or 83%). We hope that the Reviewer agrees that such speculative calculations should not be included in the paper.

3. Additional quantitative MS experiments should be included to test the changes in acetylation on each site after HDAC6 incubation.

For the various technical and conceptual reasons alluded to above, this type of MS-based quantitative assessment is fraught with interpretive difficulties.

We wish to emphasize that our goal in pursuing these MS experiments was to provide clues as to potential sites of GαqPCR-mediated channel modification, which could then be tested in follow-up functional experiments. We are pleased that this approach was effective, as those follow up experiments were successful in identifying functionally important acetyl-lysine residues on PANX1.

Minor concern:

1. *It is also important to demonstrate that PANX1 can directly interact with HDAC6 if this is not yet known.*

We believe our experiments in cell-free patches provide especially strong evidence for a functional interaction between HDAC6 and Panx1 channels. Nevertheless, we attempted to co-immunoprecipitate the enzyme and channel from HEK293T cells overexpressing a1D-AR-myc-FLAG and hPANX1, with and without PE treatment. Unfortunately, the results were inconclusive. We think this likely reflects the fact that enzyme-substrate interactions are often only transient.

Reviewer #3 (Remarks to the Author):

Deacetylation as a receptor-regulated...

By Chiu et al

In a careful and decisive manner, the authors demonstrate that HDAC6 can activate PANX1 directly by deacetylation of Lys140 and that reversible acetylation-deacetylation of the channels regulates the channel on heterologous expression. Further, the authors offer strong evidence for the operation of this non-canonical and previously unknown mechanism in T lymphocytes. The work is impressive as well for its clarity of presentation. After using pharmacology to hypothesize the mechanism, the authors use MS to find the sites of interest and then point mutation to identify the residue controlling the regulatory pathway under study. Particularly elegant, they first show the effects require a diffusible signal by a classical strategy using on-cell patch mode to isolate the channel from activator applied in the bath solution and then pull membrane patches off cells and apply the regulatory enzyme to the cytoplasmic face of the membrane to modulate the channels directly.

I have only one minor comment: in Figure 1 the blocker carbenoxolone, CBX, is used without being discussed except in the legend; this should be enunciated since it is used throughout.

We have now defined our use of the blocker carbenoxolone, CBX, in the text proper, at its first use (see **p.3**).

Beautiful work. Well done. Steve Goldstein

We are very grateful for the positive assessment of the quality and importance of this work.

Reviewers' comments:

Reviewer #1 (Remarks to the Author):

The revised manuscript by Chiu et al presents a very interesting discovery - that a specific deacetylation pathway can lead to activation of pannexin1 channels. This is a novel finding that has broad reaching implications for ion channel (and other receptor) regulation. The authors have carefully responded to my concerns from the previous review. I appreciate the inclusion of Figure R1 in the response to my concerns about a potential role of Src. However, this figure should be included in the manuscript because it is an important negative control that eliminates phosphorylation by Src at the identified Y198 and Y308 sites in the $\alpha 1R$ activation of pannexin. It is important because there are substantial data in the literature showing that Rho/Rcok-mDia can directly activate Src (see for example Tominaga et al (2000) Mol Cell 5(1)). Inclusion of Fig R1 in the final version of the paper will increase confidence in the specificity of the deacetylation model. I agree with the authors that the regulation of pannexin1 and other channels by phosphorylation is likely complex and that a detailed investigation of this is better suited for an independent study. However, I do feel strongly that it is important to include the Src kinase data in this paper.

Congratulations on an excellent and interesting study.
Roger Thompson

Reviewer #2 (Remarks to the Author):

This reviewer appreciates the authors' effort to address previous concerns. Overall, the authors have presented convincing evidence to suggest a novel regulatory pathway that involves HDAC6 and acetylation in activating PANX1 channel. However, it seems that the current data still lacks all the evidences to demonstrate the biochemical mechanism of HDAC6-mediated direct deacetylation. The manuscript would be strengthened by further analysis.

1. The authors have not been able to show that PANX1 directly interacts with HDAC6. This is reasonable given that PANX1 functions as a trans-membrane channel while HDAC6 is a cytosolic deacetylase. However, the authors have shown extensive evidence that purified HDAC6 can activate PANX1 channel on a patch membrane in a cell-free condition. It would be very difficult to understand how HDAC6 can be recruited to the cell membrane and find its specific acetyllysine targets. As the authors have performed co-immunoprecipitation but did not observe their interaction, it would be important if the authors can perform additional analysis such as immunofluorescence to demonstrate the co-localization of HDAC6 and PANX1 in cells.

2. The authors showed that the SILAC ratios for the two acetyllysine sites after normalizing with protein ratios were 0.875 and 0.912. These are very small decrease in acetylation level for quantitative proteomics analysis, typically within the range of quantification error. For confident measurement, the analysis needs to be repeated with multiple biological replicates and statistical significance test. RAW MS data for the SILAC analysis should be uploaded to publically accessible server.

3. For the deacetylation experiment with patch membrane, the KKQQ mutant study further strengthened the notion that the positive charges on K140 and K409 were critical for HDAC6-mediated channel activation. But it is not sufficient to demonstrate that the importance of HDAC6 activity in this process. A simple experiment for this purpose is to see if the treatment with HDAC6-specific chemical inhibitor can inhibit HDAC6-mediated channel activation in patch membrane. In addition, it would be important to show that site-specific acetylation levels (K140 and K409) of PANX1 on patch membrane decrease apparently after HDAC6 treatment.

Overview

We thank the Reviewers for their positive comments and helpful suggestions. We appreciate your patience as we faced some challenges with completing the experiments during the COVID pandemic. We have addressed all comments in detail below.

To briefly summarize the salient changes to the revised manuscript: 1) we were ultimately able to successfully co-immunoprecipitate HDAC6 with PANX1 to support a physical interaction between HDAC6 and PANX1 that could recruit the enzyme to the membrane-localized channel; and 2) we performed additional biological replicates of the SILAC-MS experiments that reinforced a consistent, albeit modest, deacetylation of Lys-140 after α 1D receptor stimulation. We expound on some possible technical/biological reasons for the modest effect in these biochemical experiments further below. Here, we emphasize that those MS experiments were intended to tentatively identify possible PANX1 residues for subsequent functional studies – and, for that, they proved highly effective in allowing us to make relevant channel mutations that demonstrated a direct channel activation mechanism by HDAC6-mediated deacetylation. Overall, as the Reviewers have also appreciated, we believe this work convincingly identifies a novel deacetylation mechanism for direct channel activation in a non-canonical signaling pathway by G α -coupled receptors.

Reviewer #1 (Remarks to the Author):

The revised manuscript by Chiu et al presents a very interesting discovery - that a specific deacetylation pathway can lead to activation of pannexin1 channels. This is a novel finding that has broad reaching implications for ion channel (and other receptor) regulation.

We are grateful for this assessment of the important implications of this discovery for ion channel and receptor regulation.

The authors have carefully responded to my concerns from the previous review. I appreciate the inclusion of Figure R1 in the response to my concerns about a potential role of Src. However, this figure should be included in the manuscript because it is an important negative control that eliminates phosphorylation by Src at the identified Y198 and Y308 sites in the α 1R activation of pannexin. It is important because there are substantial data in the literature showing that Rho/Rock-mDia can directly activate Src (see for example Tominaga et al (2000) Mol Cell 5(1)). Inclusion of Fig R1 in the final version of the paper will increase confidence in the specificity of the deacetylation model. I agree with the authors that the regulation of pannexin1 and other channels by phosphorylation is likely complex and that a detailed investigation of this is better suited for an independent study. However, I do feel strongly that it is important to include the Src kinase data in this paper.

We have now included the rationale that Src family kinases can be activated downstream of Rho (p. 4) and added data from the PANX1(Y198,308FF) mutated channels in Fig. S3.

Congratulations on an excellent and interesting study.

Roger Thompson

Thank you.

Reviewer #2 (Remarks to the Author):

This reviewer appreciates the authors' effort to address previous concerns. Overall, the authors have presented convincing evidence to suggest a novel regulatory pathway that involves HDAC6 and acetylation in

activating PANX1 channel. However, it seems that the current data still lacks all the evidences to demonstrate the biochemical mechanism of HDAC6-mediated direct deacetylation. The manuscript would be strengthened by further analysis.

Thank you for the positive comments and further suggestions for improvement.

1. The authors have not been able to show that PANX1 directly interacts with HDAC6. This is reasonable given that PANX1 functions as a trans-membrane channel while HDAC6 is a cytosolic deacetylase. However, the authors have shown extensive evidence that purified HDAC6 can activate PANX1 channel on a patch membrane in a cell-free condition. It would be very difficult to understand how HDAC6 can be recruited to the cell membrane and find its specific acetyllysine targets. As the authors have performed co-immunoprecipitation but did not observe their interaction, it would be important if the authors can perform additional analysis such as immunofluorescence to demonstrate the co-localization of HDAC6 and PANX1 in cells.

Thank you for suggesting the co-immunoprecipitation experiments. By using a different epitope-tag to pull down HDAC6 we have now been successful in co-immunoprecipitating PANX1 with FLAG-tagged HDAC6; these new data are presented in **Fig. 4d** (and described on **p.6** in text).

As noted, HDAC6 is a cytosolic deacetylase and it can thus access cytosolic regions of membrane-associated channels (see highlighted text, **p.5-6 & p.8**). For example, a previous study suggested that HDAC6 can deacetylate lysine residues of connexin 32 to modulate protein stability (Alaei et al., 2018, BMC Cell Biol). Another lysine deacetylase, SIRT1, has also been reported to be able to directly deacetylate Nav1.5 to regulate surface expression of the channels (Vikram et al., 2017, Nat Med). In these other cases, the deacetylase also affected membrane-localized ion channels although the effect of deacetylation was not on channel gating, as we have shown here.

2. The authors showed that the SILAC ratios for the two acetyllysine sites after normalizing with protein ratios were 0.875 and 0.912. These are very small decrease in acetylation level for quantitative proteomics analysis, typically within the range of quantification error. For confident measurement, the analysis needs to be repeated with multiple biological replicates and statistical significance test.

We have now repeated this MS analysis with two additional samples (i.e., three biological replicates of the SILAC experiment), and the data are indeed consistent in showing a modest decrease in acetylation after α 1-AR activation (particularly at K140, ~10%). The levels of K409 acetylation were variable among different preparations We now report data from these three repeat experiments in the legend to **Fig. 5a**.

We understand the Reviewer's concern and would briefly reiterate some of the points made previously that might account for the magnitude of this effect. For the MS analysis, PANX1 was immunoprecipitated from whole cell lysates while the acetylation-deacetylation may be selective only for the membrane-associated fraction of channels. (Based on the signal intensity from Western blots of cell surface biotinylation experiments, we estimate that only ~1-2% of total Panx1 protein is on the cell membrane.) Also, we do not know the stoichiometric requirements for deacetylation-mediated activation of this oligomeric channel (heptameric, as it turns out). We previously demonstrated that α 1D-AR activates Panx1 channels in a stepwise manner (Chiu et al., 2017, Nat Comm) implying that Panx1 channel activity may ensue after deacetylation on only some of the subunits of the heptameric channel.

In light of these uncertainties, we have emphasized in the text that the MS data were used to provide suggestive evidence for involvement of particular acetyl-lysine residues (e.g., **Abstract, p. 6**). Finally, although the changes were modest, we note that the MS results are presented in the context of multiple orthogonal lines of evidence that support the hypothesis that HDAC6 can mediate channel activation through a deacetylation process that likely involves K140 of PANX1.

RAW MS data for the SILAC analysis should be uploaded to publically accessible server.

The raw MS data has now been uploaded to a publically accessible server (PRIDE; Project accession: PXD025912). Reviewer may access the data using the following log-in information:

Username: reviewer_pxd025912@ebi.ac.uk

Password: G4TWPU4L

Note that we have also now included MS1 and MS2 spectra of the relevant acetylated PANX1 peptides (**Fig. S4c-f**).

3. For the deacetylation experiment with patch membrane, the KKQQ mutant study further strengthened the notion that the positive charges on K140 and K409 were critical for HDAC6-mediated channel activation. But it is not sufficient to demonstrate that the importance of HDAC6 activity in this process. A simple experiment for this purpose is to see if the treatment with HDAC6-specific chemical inhibitor can inhibit HDAC6-mediated channel activation in patch membrane.

The evidence we present that HDAC6 activity is required for channel activation includes the demonstration that catalytically-dead HDAC6 has no effect on PANX1 currents in cells (**Fig. 4b,c**) and that the effect of wild type HDAC6 requires the specific channel lysine residues K140 and K409, in both cells and in excised patches (**Fig. 5f-h,k**). Rather than test the ability of tubacin to interfere with purified HDAC6 on PANX1 isolated in membrane patches, a cell-free system, we performed additional experiments to verify that HDAC6 activity is required for PANX1 activation in cells where potential off-target or non-catalytic activity might be more likely. Thus, we now show that the HDAC6-specific inhibitor tubacin blocks effects of co-expressed HDAC6 on PANX1 channels (**Fig. 5j**).

In addition, it would be important to show that site-specific acetylation levels (K140 and K409) of PANX1 on patch membrane decrease apparently after HDAC6 treatment.

It is not technically feasible to obtain enough channel protein from a membrane patch (excised with a 1 μ m diameter pipette) to perform this analysis.

REVIEWER COMMENTS

Reviewer #2 (Remarks to the Author):

This reviewer appreciates the great efforts by the authors to further strengthen this manuscript. The new data including the interaction data, the replicate SILAC analysis and chemical inhibitor treatment experiments strongly supported a potentially novel regulatory mechanism.

One minor issue was observed in the MS/MS analysis data. The MS/MS spectrum of K140 acetylation identification in Supp fig 4c-d was a false positive identification. K140 does not localize to the C-terminus of the protein and therefore, if K140 was acetylated endogenously, trypsin would not be able to cleave the site and generate the peptide with acetylated K140 on the peptide C-terminus. Further examination of the uploaded MS identification files showed that in the 2nd repeat analysis of SILAC data, K140 acetylated peptide was indeed identified. However, this identification was not observed in the search results of the other two datasets. Given that K140 acetylation is a critical regulatory site in this study, the mass spec data should be further examined for correct identification and quantification.

1. The MS/MS spectrum for the identification of K140 acetylation should be updated in Supp fig 4d.
2. The SILAC quantification of K140 was only obtained once and replicate data should be included to demonstrate the quantification reproducibility. Supp fig 4c should be updated accordingly. In addition, please include in the Supp table, the SILAC ratios for K140 and K409 sites in replicate analysis, as well as the SILAC ratios for PANX1 protein as control. The reported SILAC ratios on K140 and K409 sites in Fig 5a should be the site ratio normalized by PANX1 protein ratio.

Overview

We appreciate the careful examination and helpful suggestions for improvement. We now present new MS/MS spectra in Supp. Figure 4c-d. We also include a new Supplementary Table showing information regarding the different hPANX1 peptides with acetylation at either Lys140 or Lys409 that were identified by SILAC-MS/MS analysis. Below please find our point-by-point responses to the Reviewer's comments.

Reviewer #1 (Remarks to the Author):

This reviewer appreciates the great efforts by the authors to further strengthen this manuscript. The new data including the interaction data, the replicate SILAC analysis and chemical inhibitor treatment experiments strongly supported a potentially novel regulatory mechanism.

We are grateful for the positive comments on the revised manuscript.

One minor issue was observed in the MS/MS analysis data. The MS/MS spectrum of K140 acetylation identification in Supp fig 4c-d was a false positive identification. K140 does not localize to the C-terminus of the protein and therefore, if K140 was acetylated endogenously, trypsin would not be able to cleave the site and generate the peptide with acetylated K140 on the peptide C-terminus. Further examination of the uploaded MS identification files showed that in the 2nd repeat analysis of SILAC data, K140 acetylated peptide was indeed identified. However, this identification was not observed in the search results of the other two datasets. Given that K140 acetylation is a critical regulatory site in this study, the mass spec data should be further examined for correct identification and quantification.

We thank the Reviewer for performing an in-depth analysis of the MS data. We understand the Reviewer's concern about false positive identifications given the location of lysine at the C-terminus in the identified peptides. We do not share this concern. We recognize that there is some debate as to whether trypsin can cleave acetylated lysine residues. However, in our experience (e.g., this dataset), we do not consider a C-terminal location of the modified Lys residues to be indicative of a "false positive". In the new Supplementary Table, we highlight the location of acetylated lysines in the identified peptides (red font) and mention that some are evident at the C-terminal tryptic digestion site (see the footnote with the Table).

We now present the spectra for the peptide identified by the Reviewer (actually from the 3rd trial) as an example of acetylated Lys140 peptide in the middle of the sequence (in the revised **Supp. Fig 4c-d**). We also corrected an oversight in the Methods, noting that we used a trypsin/Lys-C mix for digestion (p. 17).

1. The MS/MS spectrum for the identification of K140 acetylation should be updated in Supp fig 4d.

As mentioned, we replaced **Supp. Fig 4c-d** with new MS/MS spectra of an hPANX1 peptide with acetylated Lys140 in the middle of the amino acid sequence. This peptide was found twice in the third biological replicate of SILAC-MS/MS experiments, with good coverage and very high confidence scores determined by Mascot software. We would also like to note that this peptide was included in analysis of the normalized H/L ratio provided with the previous revision.

2. The SILAC quantification of K140 was only obtained once and replicate data should be included to demonstrate the quantification reproducibility. Supp fig 4c should be updated accordingly. In addition, please include in the Supp table, the SILAC ratios for K140 and K409 sites in replicate analysis, as well as the SILAC ratios for PANX1 protein as control. The reported SILAC ratios on K140 and K409 sites in Fig 5a should be the site ratio normalized by PANX1 protein ratio.

We now include a new Supplementary Table (pg. 32), in which we present raw H/L ratio of the individual peptides and the calibrated H/L ratio normalized to the overall hPANX1 peptides identified in the analysis.